

# Impact of machine learning on dietary and exercise behaviors in type 2 diabetes self-management: a systematic literature review

Rizwan Riaz Mir[1], Nazeef Ul Haq[2], Kashif Ishaq[3], Nurhizam Safie[4] and Abdul Basit Dogar[3]

[1] Department of Computer Science and IT, Virtual University of Pakistan, Lahore, Pakistan
[2] Department of Computer Science, University of Engineering and Technology Lahore, Lahore, Pakistan
[3] School of Systems and Technology, University of Management & Technology, Lahore, Pakistan
[4] Faculty of Information Science and Technology, Universiti Kebangsaan Malaysia, Malaysia

## ABSTRACT

Self-awareness and self-management in diabetes are critical as they enhance patient well-being, decrease financial burden, and alleviate strain on healthcare systems by mitigating complications and promoting healthier life expectancy. Incomplete understanding persists regarding the synergistic effects of diet and exercise on diabetes management, as existing research often isolates these factors, creating a knowledge gap in comprehending their combined influence. Current diabetes research overlooks the interplay between diet and exercise in self-management. A holistic study is crucial to mitigate complications and healthcare burdens effectively. Multi-dimensional research questions covering complete diabetic management such as publication channels for diabetic research, existing machine learning solutions, physical activity tacking existing methods, and diabetic-associated datasets are included in this research. In this study, using a proper research protocol primary research articles related to diet, exercise, datasets, and blood analysis are selected and their quality is assessed for diabetic management. This study interrelates two major dimensions of diabetes management together that are diet and exercise.

## INTRODUCTION

Diabetes is known as one of the oldest chronic diseases first reported 3,000 years back in an Egyptian manuscript. Araetus of Cappadocia (81-133 AD) first used the term "diabetes" and later in 1675 a British scientist, Thomas Willis extended the term to diabetes mellitus (diabetes: flow, mellitus: honey, sweet). In 1776, a British scientist Dobson first confirmed the excess of glucose in blood and urine streams (*Ahmed, 2002*).

Diabetes mellitus (DM) is classified into four types that are type 1 diabetes mellitus (T1DM), type 2 diabetes mellitus (T2DM), pre-diabetes, and gestational diabetes (*CDC, 2023*). Most common among the four types and approximately 90–95% of the total diabetes mellitus cases are of T2DM. When a human body is not able to use insulin well within a body due to impaired insulin secretion, insulin resistance, or both, then the

Corresponding authors
Rizwan Riaz Mir,
rizwan-mir@vu.edu.pk
Kashif Ishaq,
Kashif.ishaq@umt.edu.pk

condition is referred to as T2DM (*DeFronzo et al., 2015*). T2DM refers to the malfunctioning of beta (β) cells of the pancreas.

In 2022, 537 million of the world's population approximately was living with DM from which 360 million people were urban This is supposed to increase to 783.2 million by 2045 having 600 million people urbanized. In 2021, the mortality of 6.7 million adults (aged: 20–79) was estimated as a result of DM or its complications. Pakistan became third in the list for DM from age 20–79 were 33 Million followed by China and India in numbers that will grow to 62.2 million people diagnosed with DM in 2045 (*Sun et al., 2022*).

Diabetes also imposes a high financial cost on the country's economies, health care system, individuals with DM, and of course their families. In 2017, the average annual cost for an individual in the USA with diabetes was $16,750 (*Yang et al., 2018*).

These days, T2DM is a top of the global health concerns due to its impact on humanity in terms of physical health, cultural, social, and economic factors. Due to lifestyle, urbanization, dietary patterns, and demographic structure of the society (*Chou, Hsu & Chou, 2023*), T2DM reached alarming levels (*Sun et al., 2022*). T2DM affects almost all vital organs of the human body and if careless and unattended for a long time, results in microvascular complications including neuropathy, retinopathy, and nephropathy along with macrovascular complications like cardiovascular disorders (*Liao, Chen & Guo, 2023*; *Kata & Kaur, 2024*).

Biomarkers of blood plasma glucose concentration for diabetes are usually measured by three types of tests. These tests are hemoglobin A1c (HbA1c), fasting plasma glucose (FPG), and a 2-h oral glucose tolerance test (OGTT) (*Ortiz-Martínez et al., 2022*). Reference values for these tests according to the American Diabetes Association (ADA) and World Health Organization (WHO) are shown in Table 1 (*Genuth, Palmer & Nathan, 2021*; *World Health Organization, Management-Screening, Diagnosis and Treatment, 2016*).

It is obvious from various studies that a sedentary lifestyle along with Western-style energy-dense food (junk food and drinks) are major factors and obstacles in diabetes (especially for T2DM) prevention and management (*Kolb & Martin, 2017*). Urbanization flushes out the nomadic lifestyle. A healthy and active lifestyle minimizes the risk of diabetes to 82–89% (*Mozaffarian et al., 2009*; *Choo et al., 2021*; *Albadr et al., 2022*).

Lifestyle has five major components and dimensions that are healthy diet, exercise, sleep patterns, relaxation, and social interaction (*Kolb & Martin, 2017*). *Khosravi et al. (2021)* describe spiritual health factors in addition. List 1 contains the components of lifestyle and their measurable factors.

For diabetic patients, a healthy lifestyle is key to managing diabetes. Along with lifestyle, medication, and monitoring glucose biomarkers are also factors in the diabetes management system. Self-awareness and self-management of diabetes for patients having diabetes are so important to reduce complications and increase healthy life expectancy. It will reduce the annual financial cost for patients and their families as well as contribute positively to the country's economy and reduce the load on the healthcare system by decreasing microvascular and macrovascular complications in diabetes patients (*Kamarudin, Safie & Sallehudin, 2022*; *Sadiq et al., 2021*).

**Table 1 Diagnostic reference values for diabetes patients.** HbA1C, hemoglobin A1c; FPG, fasting plasma glucose; OGTT, 2 hour oral glucose tolerance test; ADA, American Diabetic Association; WHO, World Health Organization.

| Parameters | Normal | Pre-diabetic | Type 2 diabetic |
|---|---|---|---|
| HbA1c (by ADA) | <5.7% | 5.7–6.4% | >6.4% |
| HbA1c (by WHO) | <6.0% | 6.0–6.4% | >6.4% |
| FPG (by ADA) | <100 mg/dl | 100–125 mg/dl | >125 mg/dl |
| FPG (by WHO) | <110 mg/dl | 110–125 mg/dl | >125 mg/dl |
| OGTT | <140 mg/dl | 140–199 mg/dl | >199 mg/dl |

**List 1 Health components and measurable factors regarding diabetes.**

| Lifestyle dimensions | Measurable factors/sub-domains |
|---|---|
| Diet | Protein, Fat, Carbohydrates, Minerals, Vitamins and Fiber |
| Physical activity | Footsteps, Workouts, BMI, Heart rate, and Oxygen saturation |
| Sleep pattern | Quality Sleep Index |
| Relaxation/Mental health | Stress management |
| Social interaction | Frequency, Duration, and Intensity |

The current era is the era of technology. Although there is a current boom in research and development related to Artificial Intelligence, machine learning, and deep learning, it is obvious that healthcare professionals are not being replaced by Artificially intelligent systems. Machine and deep learning have established an impressive role in the screening, diagnosis, prediction, prevention, and management of diabetes in recent times (*Broome, Hilton & Mehta, 2020*).

## Gap within the existing review articles

As mentioned in Table 2, most of the studies done in the current field of diabetes self-management through lifestyle are focusing on one aspect or dimension of diabetes management at a time. In reality, there is an intensive relationship and dependencies between the dimensions of diabetes management. So, to avoid later complications among diabetes patients and reduce the load on the national healthcare system along with financial factors, there is a need to conduct a study that interrelates two major dimensions of diabetes management together that are diet and exercise.

## Dimensions of the gap

Dimensions are the aspects that are to be assessed during research. In the diabetes self-management system, the dimensions (mentioned in Table 2) under consideration are blood glucose levels (HbA1c, FPG, and OGTT), dietary preferences (meal timing, food preferences, food database, and calorie intake by patients), and physical lifestyle (exercise patterns and activity tracking). Along with these dimensions, we are also interested in the

**Table 2 Comparison with literature works.**

| Reference & focus of study | Quality assessment scheme | SLR | Machine learning | | Blood glucose levels | | | Dietary preferences | | | | Physical lifestyle | |
|---|---|---|---|---|---|---|---|---|---|---|---|---|---|
| | | | Machine learning | Deep learning | HbA1c | FPG | OGTT | Meal timing | Food preferences | Calorie intake | Food database | Exercise patterns | Activity tracking |
| *Abhari et al. (2019)* Nutrition Recommendation System + ML | N | Y | Y | N | Y | N | N | Y | Y | Y | N | N | N |
| *Triantafyllidis & Tsanas (2019)* ML + Health System | Y | Y | Y | Y | N | N | N | N | Y | Y | N | N | N |
| *Zhu et al. (2020)* DL + ML + Diabetes | N | Y | Y | Y | Y | Y | N | N | N | N | N | N | N |
| *Tejedor, Woldaregay & Godtliebsen (2020)* ML + Diabetes + HbA1c | N | Y | Y | N | Y | N | N | N | N | N | N | N | N |
| *Oh et al. (2021)* ML + Diet + Exercise | Y | Y | Y | N | Y | N | N | Y | Y | Y | N | Y | N |
| *Rodriguez-León et al. (2021)* ML + Diabetes + Exercise + Activity Tracking | Y | Y | Y | N | Y | N | N | N | N | N | N | Y | Y |
| *Kirk et al. (2022)* ML + Meal + Food Preferences + Calories | Y | Y | Y | N | N | N | N | Y | Y | Y | N | N | N |
| *Orue-Saiz, Kazarez & Mendez-Zorrilla (2021)* ML + Nutrition Recommendation System | Y | Y | Y | N | Y | N | N | Y | N | Y | N | N | N |
| *Makroum et al. (2022)* ML + Diabetes + Physical Lifestyle | N | Y | Y | N | Y | N | N | N | N | N | N | Y | Y |
| *Chaki et al. (2022)* ML + Diabetes + Nutrition | N | Y | Y | N | Y | Y | N | Y | N | Y | N | N | N |
| *Afsaneh et al. (2022)* ML + DL + Diabetes + Nutrition | N | N | Y | Y | Y | N | N | Y | N | Y | N | N | N |
| *Tuppad & Patil (2022)* ML + Diabetes | N | N | Y | N | Y | N | N | N | N | N | N | N | N |
| *Alsayed et al. (2023)* ML + Diabetes + Nutrition | N | Y | Y | N | Y | N | N | Y | N | Y | N | N | N |
| *Njama (2023)* ML + Diabetes | N | Y | Y | N | Y | Y | N | N | N | N | N | N | N |
| *Alanazi et al. (2023)* ML + Diabetes | N | Y | Y | N | Y | N | N | N | N | N | N | N | N |
| *Lu et al. (2023)* ML + Diabetes | N | N | Y | N | Y | Y | N | N | N | N | N | N | N |
| *Pina et al. (2023)* ML + Diabetes + Nutrition | N | N | Y | N | Y | Y | N | Y | N | Y | Y | N | N |
| This article | Y | Y | Y | - | Y | Y | Y | Y | Y | Y | Y | Y | Y |

implementation of a management system using machine learning algorithms that train systems in supervised, semi-supervised and unsupervised modes.

### Inclusion and exclusion criteria for SLRs

We have implemented several exclusion criteria for selecting studies. Firstly, we have excluded articles that are not review articles. Secondly, all those research articles are excluded that are not written in the English language. To maintain relevance to our focus, we have excluded articles that do not pertain to diabetes management through machine learning. Additionally, review articles published prior to 2019 have been excluded from our search. At last, we have excluded articles that do not specifically address diabetes management in the context of self-care. These criteria ensure that the selected articles align with our research objectives and are accessible for in-depth analysis.

### Novelty

Researchers have yet to fully understand how diet and exercise together affect diabetes management. Most studies have focused on these factors individually, leaving a gap in our knowledge of their combined impact. Addressing this gap is crucial for developing personalized self-care strategies, improving blood sugar control, and reducing the risk of complications. Using machine learning to analyze the interplay of diet and exercise can provide valuable insights that can lead to more effective diabetes management. This research has the potential to improve the overall well-being of people with diabetes.

The audience is intended for this study includes medical researchers, professionals, and academics, as well as students and educators in the field, focusing on the increasing threat of diabetic retinopathy. The review encompasses extensive details about publication information, datasets, detection methods, and performance evaluation, offering a more comprehensive and balanced approach than previous reviews.

## LITERATURE REVIEW

In this section, we will summarize the existing knowledge of the SRLs and related research articles for the advancement in the field of Diabetes management in terms of identified dimensions and sub-dimensions described in the Introduction section.

We used the following search query on Google Scholar to find related SLRs is mentioned below:

"*(Systematic Literature review) and (Diabetes) and (machine learning) and (diet) and (Exercise)*".

*Alsayed et al. (2023)* have conducted a systematic literature review that thoroughly examines the effectiveness of advanced techniques in managing diabetes self-care from 2010 to 2021. They explore a wide range of innovative tools, technologies, and strategies that go beyond traditional self-care practices. These include mobile health apps, wearable devices, telemedicine, and personalized decision support systems. The review underscores the positive impact of these advanced techniques, which lead to enhanced glycemic control, improved self-management, increased patient engagement, and personalized care.

However, it also acknowledges barriers related to cost and digital literacy that need addressing for wider adoption and equitable access to diabetes care.

*Oh et al. (2021)* discussed about artificial intelligence (AI) chatbots that are increasingly utilized to encourage physical activity, healthy dietary habits, and weight loss, as assessed in a systematic review. The majority of studies primarily focused on enhancing physical activity, with most reporting significant improvements in related behaviors. While the number of studies addressing diet and weight status was limited, two studies demonstrated notable improvements in dietary behaviors, and one observed significant weight changes post-intervention. However, the review identified challenges in AI chatbot interventions, including limited personalization, short-term follow-up, and the need for more rigorous evaluation. Despite these hurdles, the review highlights the potential of AI chatbots as valuable tools for promoting healthy lifestyles, emphasizing the necessity for further research to develop and assess personalized, long-term interventions.

*Alsayed et al. (2023)* mention machine learning (ML) and artificial intelligence (AI) which are rapidly reshaping the healthcare landscape, and diabetes care is no exception. They enable the analysis of extensive medical data, patient records, and healthcare information, unveiling patterns and insights beyond human capacity. This systematic review outlines four pivotal areas where ML and AI have a profound impact on diabetes care: automated retinopathy detection, clinical decision support, predictive population risk stratification, and self-management tools for patients. These technologies aid in early retinopathy detection, provide decision support to healthcare professionals, predict diabetes risk in individuals and populations, and offer personalized self-management tools for patients. While the review underscores the transformative potential of ML and AI in diabetes care, it also underscores the importance of addressing challenges such as the need for rigorous clinical validation, transparent and user-friendly systems, and ethical considerations related to data privacy and security to ensure their widespread adoption in clinical practice. In summary, this review provides a comprehensive overview of the current state of ML and AI in diabetes care and highlights their capacity to enhance the well-being of individuals living with diabetes.

*Makroum et al. (2022)* provide a comprehensive analysis of the utilization of machine learning and smart devices in diabetes care. They synthesize existing research and emphasize the evolving landscape of technology-driven solutions for managing diabetes. Their review offers insights into the effectiveness, user-friendliness, and potential impact of these innovative tools on healthcare. It highlights the potential of these tools in delivering personalized and real-time support to individuals with diabetes, which can result in better self-management, improved clinical outcomes, and an enhanced quality of life.

*Njama (2023)* conducts a systematic literature review on the "Adoption of Machine Learning in Diagnosis and Treatment in Healthcare." This review critically evaluates the widespread use of machine learning in the healthcare sector. It surveys a range of studies and examines the current state of machine learning applications in medical diagnosis and treatment. The review illuminates the significant progress made in integrating machine learning algorithms, discussing their potential to enhance diagnostic accuracy, treatment recommendations, and overall patient care. It also identifies key challenges such as data

privacy and interpretability, emphasizing the need for further research and standardization to fully unleash the potential of machine learning in healthcare settings.

*Chaki et al. (2022)* offer a comprehensive assessment of the application of machine learning and artificial intelligence in detecting and self-managing diabetes mellitus. They synthesize a wide range of studies to provide insights into the advancements and effectiveness of these technologies in diabetes care. The review highlights the potential of machine learning and AI in accurately diagnosing diabetes, personalizing treatment, and assisting individuals in self-managing their condition, ultimately leading to improved patient outcomes and healthcare delivery. The need for further research and the integration of these technologies to optimize diabetes care in the future is also discussed.

*Alanazi et al. (2023)* provide a comprehensive analysis of the utilization of machine learning and AI in the context of diabetes management. This review synthesizes a wide range of research studies and underscores the growing importance of these technologies in enhancing diabetes care. It highlights their potential to improve early diagnosis, predict complications, optimize treatment strategies, and support self-management for individuals with diabetes. The review also examines the current challenges and limitations, such as data privacy concerns and the need for standardized approaches, while emphasizing the promising role of machine learning and AI in shaping the future of diabetes care and outcomes.

*Rodriguez-León et al. (2021)* conducted a comprehensive examination of the use of mobile and wearable devices in monitoring various parameters related to diabetes. Their review synthesizes a diverse body of research to provide insights into the evolving landscape of technology-driven solutions for diabetes management. The review highlights the effectiveness of these tools in tracking glucose levels, physical activity, and other relevant parameters, ultimately enabling individuals with diabetes to better manage their condition. The review also addresses the challenges, such as data accuracy and usability, and underscores the potential of mobile and wearable technology in revolutionizing the monitoring and self-management of diabetes, promising to enhance overall health outcomes for patients.

*Lu et al. (2023)* delve into the promising intersection of digital health and machine learning in the context of gestational diabetes care. This systematic review synthesizes a range of studies, shedding light on the utilization of technology to monitor and manage blood glucose levels in pregnant women with diabetes. It underscores the potential for innovative digital solutions and machine learning algorithms to provide real-time, personalized support, ultimately improving the outcomes and well-being of expectant mothers with gestational diabetes. Additionally, the review addresses the current state of research and highlights the need for continued innovation and research in this critical area of healthcare.

*Afsaneh et al. (2022)* offer a detailed analysis of the evolving landscape of artificial intelligence in diabetes care in their comprehensive review titled "Recent Applications of Machine Learning and Deep Learning Models in the Prediction, Diagnosis, and Management of Diabetes." They synthesize a wide array of studies and emphasize the increasing significance of machine learning and deep learning in enhancing various aspects

of diabetes management. The review highlights the applications of these models in predicting diabetes risk, improving diagnostic accuracy, and personalizing treatment plans. It underscores their potential to revolutionize healthcare delivery and patient outcomes by offering data-driven, precise, and proactive approaches to diabetes care. The review also underscores the need for ongoing research and validation to ensure these advanced technologies are effectively integrated into clinical practice.

*Pina et al. (2023)* explore the fusion of large-scale data analytics and machine learning to address the multifaceted challenges of diabetes care. Their review highlights the transformative potential of leveraging extensive healthcare data to improve diabetes management, from early detection and risk prediction to tailoring treatment plans and monitoring patient progress. By harnessing the power of data-driven insights and machine learning algorithms, this approach holds the promise of more personalized, efficient, and effective care for individuals with diabetes, ultimately leading to enhanced health outcomes and better management of this chronic condition. The review underscores the importance of continued research and collaboration between healthcare, data science, and machine learning experts to fully realize the potential of big data in diabetes management.

*Zhu et al. (2020)* describe a comprehensive assessment of the applications of deep learning in the realm of diabetes care. This review synthesizes a wide range of research studies, highlighting the growing significance of deep learning in various aspects of diabetes management, including diagnosis, risk prediction, glucose monitoring, and treatment personalization. It underscores the potential of deep learning models to provide more accurate and robust solutions for addressing the complexities of diabetes, ultimately leading to better clinical outcomes and improved patient care. The review also addresses the challenges and limitations in this field, emphasizing the need for further research and development to harness the full capabilities of deep learning in diabetes care.

Nutrition recommendation systems (NRS) are like digital nutrition coaches that offer personalized healthy eating tips, catering to various goals like weight loss, managing health conditions, or just eating better. This detailed review dived into the technical aspects of these systems. It found that most NRSs use a clever combo of recommendation techniques, such as analyzing what you like, what others with similar tastes prefer, and general dietary guidelines, to give you more precise advice. They also need to store a lot of nutritional info, like what's in different foods and what's good for you, using various methods like organized lists or databases. Additionally, NRS employs complex math to create your personalized tips, factoring in your health goals and food preferences. And to make sure they actually help you, they get tested through things like user feedback, tracking what you eat, and even clinical trials. So, while NRSs have the potential to be your nutrition buddy, more research is needed to fine-tune them for accuracy and make sure they truly help you achieve your dietary goals (*Abhari et al., 2019*).

The systematic review by *Triantafyllidis & Tsanas (2019)* delved into the realm of ML in digital health interventions, exploring eight real-life studies targeting various health conditions. These studies employed a range of ML algorithms, including decision trees, support vector machines, and deep learning models, with mixed but promising outcomes. Notably, an ML-powered intervention for depression showed effectiveness in reducing

depressive symptoms, while a smoking cessation intervention using ML was successful in increasing cessation rates. Challenges unveiled in the review included the demand for ample high-quality data for ML training, the necessity for personalized ML algorithms, and the complexity of evaluating these interventions due to the intricate nature of human behavior and ethical considerations. In essence, the review suggests that ML holds significant potential for crafting effective digital health interventions, but calls for further research to address the associated challenges.

ML is poised to revolutionize the way doctors make decisions about diabetes care. ML algorithms can analyze vast amounts of medical data to identify patterns and insights that would be impossible for humans to see on their own. This can help doctors to better predict which patients are at risk of developing complications, tailor treatment plans to individual needs, and optimize insulin dosing. However, the review also underscores several challenges that warrant attention before ML-driven CDSS can be seamlessly incorporated into clinical practice. These challenges encompass the necessity for more rigorous clinical trials to validate the efficacy of ML algorithms, the development of transparent and user-friendly systems, and addressing ethical concerns pertaining to data privacy and security. In conclusion, while the review paints a promising picture of ML's potential to substantially enhance diabetes CDSS, further research is crucial to surmount the highlighted challenges. Several real-world instances demonstrate how ML is already enhancing diabetes CDSS, from predicting diabetes risk with remarkable accuracy to optimizing insulin dosing and offering personalized guidance to patients based on their real-time blood sugar data. As ML technology advances, we can anticipate even more innovative and effective applications to support diabetes management (*Tuppad & Patil, 2022*).

Precision nutrition is an individualized dietary approach considering one's unique genetic, metabolic, and environmental factors to provide personalized dietary guidance for health and wellness goals. A systematic review highlights its potential to improve various health outcomes, including weight management, blood sugar control, cardiovascular health, cognitive function, and more. It also shows promise in assisting individuals with chronic conditions like diabetes and cancer. Key areas of precision nutrition research encompass nutrigenomics, examining genetic impacts on food responses, metabolomics to identify dietary response biomarkers, and foodomics to analyze food composition. Despite its promise, challenges such as cost, complexity, and the need for more evidence must be addressed for wider adoption. Current applications include direct-to-consumer genetic testing and healthcare programs for managing chronic conditions. As research advances, precision nutrition may offer innovative ways to enhance human health (*Kirk et al., 2022*).

NRS are computer-based tools that offer personalized guidance for healthier eating, addressing diverse goals like weight management, chronic disease care, and overall well-being. In a systematic review spanning studies from 2012 to 2020, 25 NRS were examined, utilizing various methods such as content-based filtering, collaborative filtering, rule-based filtering, and hybrid approaches. NRS demonstrated their effectiveness in enhancing dietary quality by suggesting nutrient-rich, low-fat, low-sugar, and low-salt food choices.

They also proved valuable in facilitating weight loss through individualized calorie and macronutrient recommendations, aiding blood sugar control with guidance toward low-carb, high-fiber meals, and supporting cardiovascular health by encouraging diets low in saturated fat and cholesterol while rich in fiber. Challenges in NRS development and assessment involve the complexities of personalization, accuracy, and user-friendliness. While NRS exhibits great potential for improving nutrition and health, more research is essential to overcome these challenges. Current applications encompass mobile apps that offer personalized meal plans and recipes, websites generating customized shopping lists, and healthcare providers delivering personalized dietary counseling. As NRS technology advances, we can anticipate the emergence of even more innovative and effective approaches to enhance human well-being (*Orue-Saiz, Kazarez & Mendez-Zorrilla, 2021*).

This comprehensive review on the use of reinforcement learning (RL) in managing blood glucose levels in individuals with diabetes analyzed 20 studies spanning from 2013 to 2023, examining various RL-based systems for blood glucose control by *Tejedor, Woldaregay & Godtliebsen (2020)*. The findings indicated that RL-based approaches can significantly improve blood sugar management in diabetes patients. However, the review also underscored the critical challenges that need to be overcome for wider adoption, such as the requirement for more rigorous clinical trials, the development of secure and dependable systems to prevent issues like hypoglycemia, and the creation of user-friendly solutions for both patients and healthcare professionals. In summary, the review suggests that RL-based blood glucose control systems have the potential to revolutionize diabetes care, but additional research is essential to address these challenges. Notable examples of RL applications include a University of California, Berkeley clinical trial involving an RL-based insulin pump control system, the RL-based system by Medtronic available for managing type 2 diabetes, and the forthcoming RL-based insulin pump control system by Tandem Diabetes Care for type 1 diabetes patients. With the ongoing advancement of RL technology, we can anticipate the emergence of more innovative and effective approaches to enhance diabetes management and enhance the well-being of those living with the condition.

These reviews collectively underscore the growing significance of machine learning, artificial intelligence, and data analytics in diabetes management. They highlight the potential benefits and challenges associated with these technologies in improving patient outcomes and healthcare delivery. Short summaries and limitations of the SLRs are also written below in Table 3.

The current review has yet to investigate the combined impact of diet and exercise in the context of diabetes management with the help of computer-aided intelligence. Recognizing this gap, there is a pressing need for a substantial research undertaking. The study in question is composed to explore the interconnected effects of dietary and physical activity factors on self-directed diabetes management. It aims to leverage the potential of supervised machine learning methods to proactively prevent diabetic complications. By venturing into this unexplored domain, the research is to provide some insights into the complicated relationship between diet and exercise within diabetes self-care.

**Table 3  Short summaries and limitations of literature work.**

| Paper | Short summary | Limitations |
|---|---|---|
| Alsayed et al. (2023) | Alsayed et al. (2023) conducted a comprehensive systematic review of advanced diabetes self-care techniques from 2010 to 2021, showing their positive impact on glycemic control, self-management, and patient engagement while addressing cost and digital literacy barriers for broader adoption. | Short follow-up period Publication Bias |
| Makroum et al. (2022) | Makroum et al. (2022) offer a comprehensive analysis of machine learning and smart devices in diabetes care, emphasizing their potential for personalized support, better self-management, improved clinical outcomes, and enhanced quality of life. | Data Quality System Transparency |
| Njama (2023) | Njama (2023) critically assesses the adoption of machine learning in healthcare, highlighting its potential for improving diagnostic accuracy and treatment recommendations while identifying challenges like data privacy and the need for more research and standardization. | Focus on high-income countries Publication Bias |
| Chaki et al. (2022) | Chaki et al. (2022) offer a comprehensive assessment of machine learning and AI in diabetes care, emphasizing their potential for accurate diagnosis, personalized treatment, and enhanced self-management, while emphasizing the importance of further research and integration for future optimization. | Heterogeneity of research |
| Alanazi et al. (2023) | Alanazi et al. (2023) provide a comprehensive analysis of machine learning and AI in diabetes management, emphasizing their potential for early diagnosis, complication prediction, treatment optimization, and self-management support, while addressing challenges like data privacy and the need for standardized approaches, shaping the future of diabetes care and outcomes. | Data Extraction Bias Publication Bias |
| Rodriguez-León et al. (2021) | Rodriguez-León et al. (2021) conducted a comprehensive examination of mobile and wearable devices in diabetes management, emphasizing their effectiveness in tracking glucose levels and physical activity, while also addressing challenges like data accuracy, and highlighting the potential to revolutionize monitoring and self-management for improved diabetes outcomes. | Data Extraction and Reporting bias Timeframe Limitation |
| Lu et al. (2023) | Lu et al. (2023) explore the intersection of digital health and machine learning in gestational diabetes care, emphasizing the potential for real-time, personalized support and improved outcomes for expectant mothers, while underscoring the need for continued innovation and research in this area. | Short follow-up Period Lack of Control Group |
| Afsaneh et al. (2022) | Afsaneh et al. (2022) provide an in-depth analysis of the growing role of machine learning and deep learning in diabetes care, emphasizing their applications in predicting risk, enhancing diagnostics, and personalizing treatment plans, with the potential to revolutionize healthcare through data-driven, precise, and proactive approaches while also stressing the importance of ongoing research and validation for effective clinical integration. | Data Quality |
| Pina et al. (2023) | Pina et al. (2023) explore the integration of large-scale data analytics and machine learning in diabetes care, emphasizing the potential for more personalized, efficient, and effective management through early detection, risk prediction, tailored treatment plans, and progress monitoring while underlining the need for ongoing research and interdisciplinary collaboration to harness the full potential of big data in diabetes care. | Publication Bias |
| Zhu et al. (2020) | Zhu et al. (2020) present a comprehensive assessment of deep learning in diabetes care, emphasizing its growing significance in diagnosis, risk prediction, glucose monitoring, and treatment personalization, with the potential for more accurate solutions, while also highlighting challenges and the need for further research to maximize its capabilities in diabetes management. | Difficult to interpret Model Working to predict Timeframe Limitation |

# SURVEY METHODOLOGY

We must have some guidelines to reduce the factor of research biases. Therefore, research protocol guidelines must be listed to reduce factors of biases. Brereton et al. (2007) and Kitchenham (2004) suggested guidelines to establish a research protocol for the Software Engineering field. We are going to follow these guidelines for our systematic literature

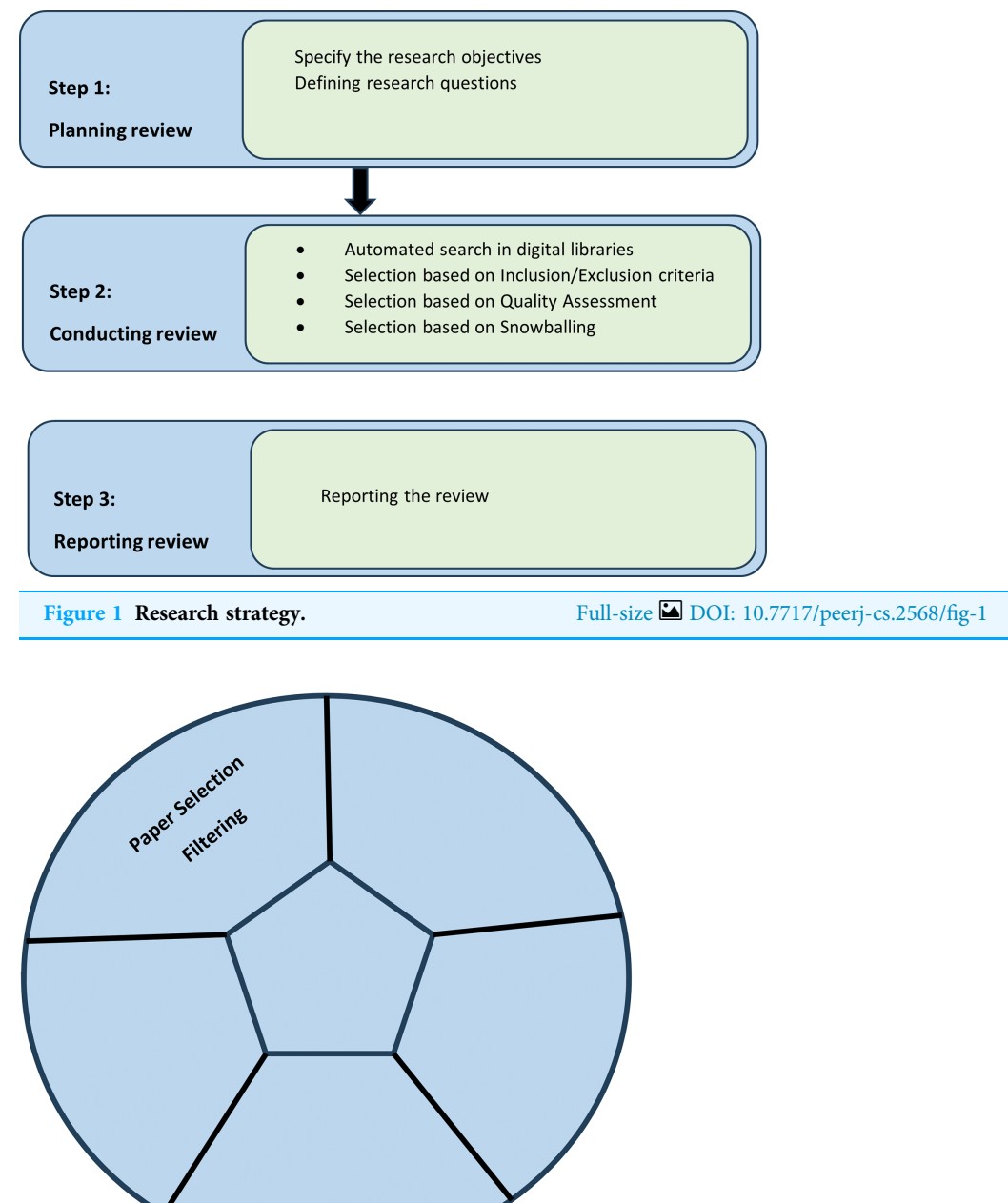

**Figure 1   Research strategy.**     

**Figure 2   Research methodology.**     

review. There are three stages in research protocol which are plan, conduct, and review presented in Fig. 1, which we followed in our systematic literature review research methodology.

## Review plan

A well-structured search strategy has been devised to identify all pertinent studies. Figures 1 and 2 illustrate the research methodology, depicting the search procedure for

**Table 4 Research questions, objectives and motivations.**

| RQs | Research question statement | Objectives and motivation |
|-----|-----------------------------|---------------------------|
| RQ1 | Which are the relevant publication channels for diabetes self-management? Which channel types and geographical areas target diabetes self-management? | To identify<br>- High quality publications venue for diabetes management research<br>- Diabetes management research published during Jan 2019 till November 2023<br>- Scientometric analysis based on meta information of the selected articles including research type, approaches, and validation methods. |
| RQ2 | What existing solutions are available for diabetes self-management for diabetes patients through machine learning? | - Evaluate the effectiveness of supervised machine learning-based solutions in aiding diabetes patients with self-management.<br>- Identify the best practices and techniques employed in current supervised machine learning solutions for diabetes self-management. |
| RQ3 | What techniques or methods are available for patient diabetes management by physical activity tracking and diet through machine learning or deep learning? | - Assess the performance and accuracy of supervised machine learning algorithms in tracking and analyzing physical activity data for diabetes management.<br>- Explore ways to optimize and personalize diabetes management strategies based on the data collected through supervised machine learning. |
| RQ4 | What datasets are mostly used to manage diabetes through lifestyle management in terms of physical activity and diet? | - Assess and identify the most frequently utilized datasets for tracking physical activity and dietary habits in individuals with diabetes.<br>- Investigate methods for integrating and analyzing these datasets to derive valuable insights for diabetes management.<br>- Explore the correlation between the data from these datasets and diabetes management outcomes. |

relevant publications, the establishment of a classification system, and the mapping of these publications. This review has rigorously adhered to a highly organized process, which included:

- Research objectives
- Device research questions
- Organize searches of databases
- Selection of studies
- Screening of relevant studies
- Data extraction
- Result combining
- Finalizing the review report

To achieve these research objectives, we have finalized our research questions and their motivation based on Table 4. The research questions are:

*RQ1:* Which are the relevant publication channels for diabetes self-management? Which channel types and geographical areas target diabetes self-management?

*RQ2*: What existing solutions are available for diabetes self-management for diabetes patients through machine learning or deep learning?

*RQ3*: What techniques or methods are available for patient diabetes management by physical activity tracking and diet through machine learning or deep learning?

*RQ4*: What datasets are mostly used to manage diabetes through lifestyle management in terms of physical activity and diet?

## Review conduct

To carry out this evaluation, we proceeded through a series of four steps. Initially, we scoured popular digital libraries for pertinent primary studies. Next, we made our selections following predetermined inclusion and exclusion criteria. Subsequently, we developed quality assessment criteria to enhance the overall review quality. Lastly, we engaged in backward snowballing to pinpoint additional significant candidate articles.

### Automated search result in digital libraries

To find relevant studies and avoid irrelevant ones, we conducted a systematic search. We used both automated and manual search methods to find studies that matched our search terms. We searched multiple digital libraries, choosing ones that are often used for systematic literature reviews. We also included widely used public platforms that are relevant to our review. Additionally, we used Google Scholar, which has resources that we did not search directly in our survey. In total, we chose ten digital sources as our primary search outlets for automated searching. These sources likely cover nearly all relevant research. These ten digital resources are:

- ACM Digital Library (http://dl.acm.org)
- IEEE eXplore (http://ieeexplore.ieee.org)
- PLOS ONE (https://journals.plos.org/plosone/)
- ScienceDirect (https://www.sciencedirect.com)
- SpringerLink (https://link.springer.com/)
- WileyOnlineLibrary (https://onlinelibrary.wiley.com/)
- arXiv (https://arxiv.org/search/cs)
- AIS eLibrary (https://aisel.aisnet.org/)
- IGI Global (https://www.igi-global.com/search/)
- Google Scholar (https://scholar.google.com/)

We manually searched for additional studies on diabetes self-management and supervised machine learning. To make the search results more relevant, we:

- Identified primary keywords based on our research questions.
- Added secondary keywords and synonyms to the primary keywords.

**List 2  Search query.**

("Deep Learning" OR "Machine Learning" OR "Artificial Intelligence")

AND

("Diabetes" OR "Diabetes Mellitus")

AND

("Recommender System" OR "Management System" OR "Self" OR "Self-Care")

AND

("Diet" OR "Nutrition" OR "Meal" OR "Food")

AND

("Exercise" OR "Activity")

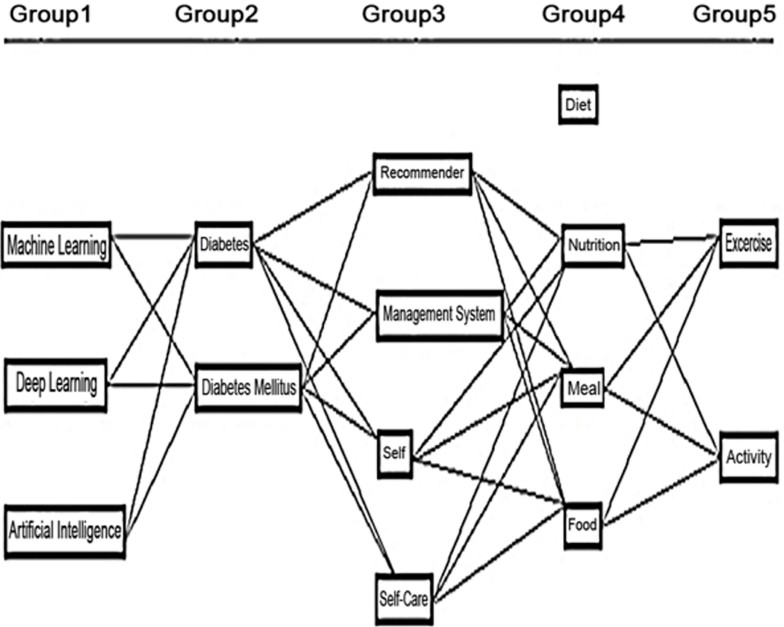

**Figure 3  Search strings used to describe work included in our Knowledge Base.**

- Used Boolean operators 'AND' and 'OR' to create a comprehensive search string.

List 2 describes the sample search query to find articles on digital libraries. Primary and secondary (synonyms) keywords as the most important words in the search string and combine them with any relevant group to form a search query. Examples of the search string on the basis of the search query are shown in Fig. 3. Boolean operators are also used to make the search more comprehensive.

**Table 5 Search strategy for digital libraries.**

| Digital libraries | Search query | Applied filter |
|---|---|---|
| ACM | [[All: Deep Learning] OR [All: Machine Learning] OR [Artificial Intelligence]] AND [[All: Diabetes] OR [All: Diabetes Mellitus]] AND [[All: Recommender System] OR [All: Management System] OR [All: Self] OR [All: Self-Care]] AND [[All: Diet] OR [All: Nutrition] OR [All: Meal] OR [All: Food]] AND [[All: Exercise] OR [All: Activity]] | [(01/01/2019 TO 11/05/2023)] |
| IEEExplore | (Deep Learning OR Machine Learning OR Artificial Intelligence) AND (Diabetes OR Diabetes Mellitus) AND (Recommender System OR Management System OR Self Care) AND (Diet OR Nutrition OR Meal OR Food) AND (Exercise OR Activity) | 2019–2023 |
| PLOS ONE | "(Deep Learning OR Machine Learning OR Artificial Intelligence) AND (Diabetes OR Diabetes Mellitus) AND (Recommender System OR Management System OR Self Care) AND (Diet OR Nutrition OR Meal OR Food) AND (Exercise OR Activity)" | Jan 1, 2019 TO Nov 5, 2023 |
| Google scholar | (Deep Learning OR Machine Learning OR Artificial Intelligence) AND (Diabetes OR Diabetes Mellitus) AND (Recommender System OR Management System OR Self Care) AND (Diet OR Nutrition OR Meal OR Food) AND (Exercise OR Activity) | 2019–2023 |
| ScienceDirect | (Deep Learning OR Machine Learning OR Artificial Intelligence) AND (Diabetes OR Diabetes Mellitus) AND (Recommender System OR Management System OR Self Care) AND (Diet OR Nutrition OR Meal OR Food) AND (Exercise OR Activity) | 2019–2023 |
| SpringerLink | '(Deep Learning OR Machine Learning OR Artificial Intelligence) AND (Diabetes OR Diabetes Mellitus) AND (Recommender System OR Management System OR Self Care) AND (Diet OR Nutrition OR Meal OR Food) AND (Exercise OR Activity)' | Computer Science, 2019–2023 |
| Wiley online | "(Deep Learning OR Machine Learning OR Artificial Intelligence) AND (Diabetes OR Diabetes Mellitus) AND (Recommender System OR Management System OR Self Care) AND (Diet OR Nutrition OR Meal OR Food) AND (Exercise OR Activity)" | 2019–2023 |
| arXiv | (Deep Learning OR Machine Learning OR Artificial Intelligence) AND (Diabetes OR Diabetes Mellitus) AND (Recommender System OR Management System OR Self Care) AND (Diet OR Nutrition OR Meal OR Food) AND (Exercise OR Activity) | Computer Science, 2019–2023 |
| Elsevier | (Deep Learning OR Machine Learning OR Artificial Intelligence) AND (Diabetes OR Diabetes Mellitus) AND (Recommender System OR Management System OR Self Care) AND (Diet OR Nutrition OR Meal OR Food) AND (Exercise OR Activity) | 2019–2023 |
| IGI Global | (Machine Learning) AND (Diabetes) AND (Self-care Management System) | Individual Journal Articles (2019–2023) |

However, our initial search query was too restrictive. It did not find articles that were only about Diabetes Self-Management. To fix this, search strings are refined for ten digital libraries by using specific filters and limiting the number of keywords. For ACM journals, ScienceDirect, and PLOS ONE, a search was done in the titles of the articles. Digital libraries were searched in all fields because they didn't allow to search more specifically. The search string was also modified for the IGI Global journal to include fewer keywords, as shown in Table 5.

*Selection based on inclusion/exclusion criteria*
**Inclusion criteria:**

i. The included article must be from the relevant domain of diabetes self-management in terms of diet and exercise through machine learning.

ii. Articles must address research questions provided in Table 3 in terms of their objectives and motivation.

iii. Only journals and reputed conferences and workshop articles must be included.

**Exclusion criteria:**

i. Exclude articles that are not written in the English language.

ii. Exclude articles that are not discussing diabetes management through machine learning.

iii. Articles before 2019 are excluded from the search.

iv. Articles that do not discuss diabetes management in terms of self-care are excluded.

v. Articles that do not discuss self-care in terms of diet and/or exercise are excluded.

vi. Different relevant articles by the same group of authors in the same domain are excluded.

vii. Most relevant and recent are kept if written by the same author or group of authors.

### Selection based on quality assessment

The critical step in conducting any review is the selection of appropriate and relevant studies based on a quality assessment (QA) process. Given the various designs of primary studies, to carry out QA in our review, have adopted quantitative, qualitative, and mixed-method critical appraisal tools recommended by *Fernandez, Insfran & Abrahão (2011)* and *Ouhbi et al. (2015)*.

To enhance the robustness of our study, we introduced a customized questionnaire to evaluate the quality of the chosen articles. Each study was evaluated using the following criteria:

a. A study received a score of (2) if it contributed to the aspect of diabetes management; otherwise, it was scored as (0).

b. A study received a score of (2) if it contributed to the aspect of machine learning/deep learning algorithms/techniques; otherwise, it was scored as (0).

c. A study received a score of (2) if it contributed to the aspect of diabetes management in terms of diet and exercise; diabetes management in terms of diet or exercise (1) or otherwise, it was scored as (0).

d. Regarding the use of the dataset, studies were assessed as either "Public Dataset (2)," "Dataset not available Publically (1)," or "No Dataset is being used or Dataset not available publically (0)."

e. Regarding the empirical type of study as either "Case Study (2)," "Review (1)," or "All Other Types (0)."

f. Studies presenting results as research type is applied then were awarded a score of (1), while those that did not were scored as (0).

g. To rate the studies, we considered factors such as computer science conference rankings (*CORE, 2023*), journal quality, and country ranking lists (*Scimago Lab, 2023*). The potential scores for publications from reputable and established sources are detailed in Table 6.

**Table 6 Possible ratings for recognized and stable publication source.**

| Publication sources | +4 | +3 | +2 | +1 | +0 |
|---|---|---|---|---|---|
| Journals | Q1 | Q2 | Q3 | Q4 | No JCR ranking |
| Conferences | Core A* | Core A | Core B | Core C | Not in core ranking |
| Workshops | Core A* | Core A | Core B | Core C | Not in core ranking |

A final score was calculated for each study by summing the scores from the above questions, resulting in a number ranging from 0 to 15. Articles that achieved scores of 8 or higher were included in the final results.

### Selection based on snowballing

After the selection of articles on the basis of performing a quality assessment, snowballing is performed by following the reference list of finalized articles. Only those studies are shortlisted and finalized which passes the inclusion/exclusion criteria. The articles were selected after reading its abstract at first glance, then selection is performed on the basis of reading introduction and conclusion and finally on the basis of critical review of the selected article. Through snowballing, (*Davis et al., 2020*), (*Preum et al., 2021*), and (*Koh-Banerjee et al., 2004*) article are selected from *Chowdhury et al. (2023)* article after passing the inclusion and exclusion criteria.

## Review report

An overview of selected studies is provided in this section.

### Overview of intermediate selection process outcome

The dynamic field of diabetes self-management and machine learning is currently packed with alive activity. Within the constant flow of new research, discriminating the most valuable studies has proven to be a difficult challenge. Thus, we get on an in-depth hunt to explore the most relevant studies, scouring through ten premier digital libraries. Our endeavor cast a wide net, encompassing a staggering 87,000 articles published since 2019.

Confronted by the overwhelming area of data, we recognized the need for precision in our search. Methodically studying titles, and abstracts, and delving into full articles as necessary, we selected through the extensive collection. Articles of lesser relevance or shortness were set aside, resulting in a more manageable collection of potential research articles.

Our next step involves delving into the most valued publications within the domain of diabetes self-management. Through a careful examination, we aim to construct knowledge, and collect articles that represent radical insights and realistic solutions. Prioritizing studies that genuinely contribute to the field, our aspiration is to unravel the secrets that empower individuals with diabetes and seamless condition of self-management using AI, or machine learning, or deep learning.

**Table 7 Selection phase and filtering.**

| Phase | Selection | Criteria | ACM digital library | IEEExplorer | PLOS ONE | ScienceDirect | SpringerLink | Wiley online library | arXiv | Elsevier | IGI global | Google Scholar | Total articles |
|---|---|---|---|---|---|---|---|---|---|---|---|---|---|
| 1 | Search | Keywords Search (Table 4) | 36,760 | 25,287 | 1,056 | 2,875 | 579 | 1,237 | 570 | 973 | 1,025 | 17,200 | 87562S |
| 2 | Filtering | Title | 348 | 201 | 153 | 279 | 197 | 261 | 207 | 178 | 248 | 532 | 2,604 |
| 3 | Filtering | Abstract | 15 | 18 | 10 | 12 | 34 | 17 | 7 | 21 | 7 | 348 | 489 |
| 4 | Filtering | Introduction and conclusion | 10 | 10 | 3 | 5 | 14 | 5 | – | 10 | – | 198 | 255 |
| 5 | Critical read | Full article | – | 8 | 1 | – | 12 | 2 | – | 7 | – | 40 | 70 |

### Overview of selected studies

After leaping into a massive pool of research across ten digital libraries, we emerged with a curated selection of 70 articles. Our initial automated search pulled in countless studies, but meticulous filtering and inspection phases helped us hone in on the gems that truly shine. Table 7 captures the exciting results of this journey, showcasing the most valuable contributions to the field.

## Assessment and discussion of research questions

In this section, we analyzed the finalized 70 primary studies based on our research questions.

### Assessment of RQ1: Which are the relevant publication channels for diabetes self-management? Which channel types and geographical areas target diabetes self-management?

Analyzing diabetes, machine learning tools, methods, and lifestyle factors poses a significant challenge for researchers engaged in developing recommendation systems for diabetes patients. To address this challenge, it is imperative to identify high-quality publication venues and conduct an analysis based on meta information within this domain. This section provides an insightful understanding of publication sources, types, years, geographical distribution, and the publication channel-wise distribution of selected studies in the field of diabetes management and machine learning research.

Following the inspection phase, a maximum of 12 studies has been finalized from the Springer digital library, and a nearly equal number has been selected from IEEE Xplore, as illustrated in Table 7. This underscores the significance of these publication sources, as both Springer and IEEE Xplore are recognized as the world's largest professional societies, publishing over 50 scholarly peer-reviewed journals across various computing and information technology disciplines.

Figure 4 represents the number of publications with respect to year of publication. Around 43% of the selected studies come from the years 2022 and 2023, indicating

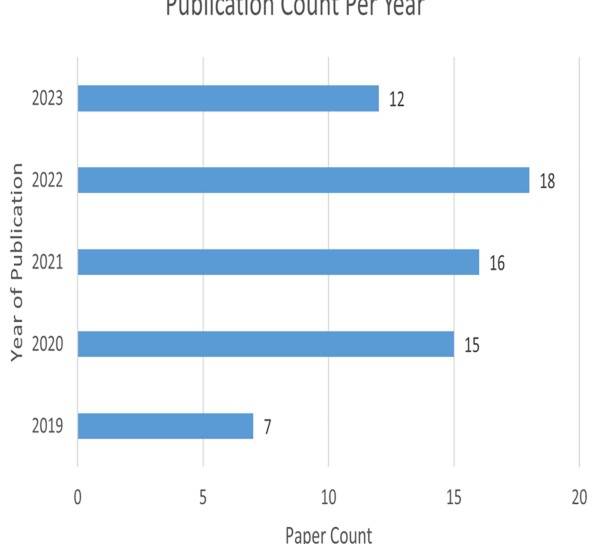

**Figure 4 Publication count per year.**

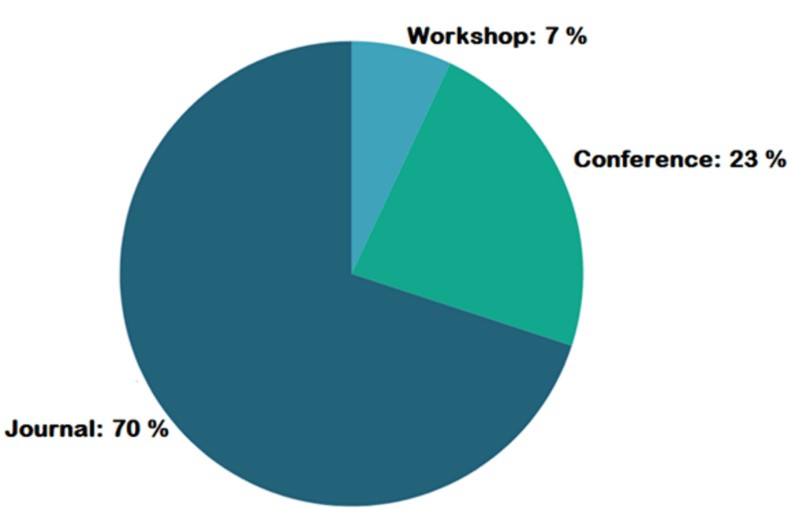

**Figure 5 Publication type geographic distribution.**

increased researcher interest in those years. As seen in Fig. 5, most of the studies were published in reputable journals, followed by ranked conferences, with only a few studies coming from workshops. Figure 6 highlights the geographical distribution, showing that the majority of publications are from various countries in America and Europe. Remaining publications have been selected from other geographical areas to reduce biases. We tried to select only reputed journal publications including all geographical areas.

The comprehensive classification outcomes and quality assessment of the concluded studies are detailed in Table 7. The studies were systematically classified according to five pivotal factors, namely research type, empirical type, and methodology. Our categorization of research types encompasses systematic literature review (SLR), solution proposals, evaluation research, experience articles, frameworks, and reviews. This structured

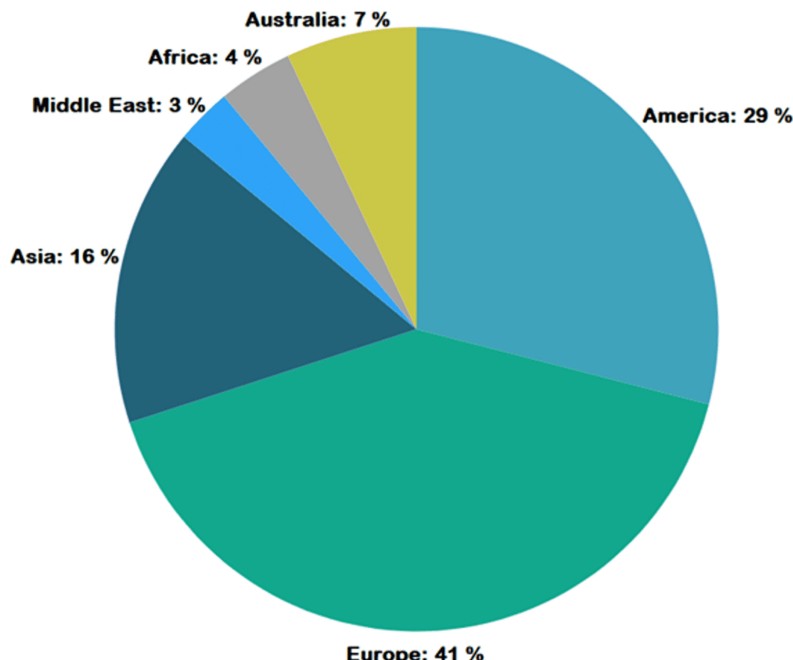

**Figure 6 Geographic distribution of publications.**

| Table 8 Quality assessment score for selected articles. | | | | | | | | | | | | |
|---|---|---|---|---|---|---|---|---|---|---|---|---|
| Citation | Publication type | P. Year | Research type | Empirical type | a | b | c | d | e | f | g | Score |
| Sowah et al. (2020) | Journal Article | 2020 | Applied Research | Case Study | 2 | 2 | 0 | 2 | 2 | 1 | 4 | 13 |
| Ngo et al. (2019) | Journal Article | 2019 | Applied Research | Case Study | 2 | 2 | 2 | 2 | 0 | 1 | 4 | 13 |
| Omisore et al. (2021) | Journal Article | 2021 | Applied Research | Case Study | 2 | 2 | 1 | 2 | 1 | 1 | 3 | 12 |
| Joachim et al. (2022) | Journal Article | 2022 | Applied Research | Case Study | 2 | 2 | 2 | 2 | 2 | 1 | 4 | 15 |
| Mogaveera, Mathur & Waghela (2021) | Conference article | 2021 | Applied Research | Case Study | 2 | 2 | 2 | 2 | 2 | 1 | 3 | 14 |
| Iwendi et al. (2020) | Journal Article | 2020 | Applied Research | Case Study | 2 | 2 | 2 | 2 | 2 | 1 | 4 | 15 |
| Tabassum et al. (2021) | Journal Article | 2021 | Applied Research | Case Study | 2 | 2 | 2 | 2 | 2 | 1 | 2 | 13 |
| Nagaraj et al. (2022) | Conference Paper | 2022 | Applied Research | Case Study | 2 | 2 | 2 | 2 | 2 | 1 | 2 | 13 |
| Zhu et al. (2022a) | Journal Article | 2022 | Applied Research | Case Study | 2 | 2 | 2 | 2 | 2 | 1 | 3 | 14 |
| Sajid et al. (2022) | Journal Article | 2022 | Applied Research | Case Study | 2 | 2 | 2 | 2 | 2 | 1 | 4 | 15 |
| Faruqui et al. (2019) | Journal Article | 2019 | Applied Research | Case Study | 2 | 2 | 2 | 2 | 2 | 1 | 4 | 15 |
| Geetha et al. (2021) | Conference Paper | 2021 | Applied Research | Case Study | 2 | 2 | 2 | 2 | 2 | 1 | 3 | 14 |
| Oka et al. (2019) | Journal Article | 2019 | Applied Research | Case Study | 2 | 2 | 2 | 2 | 2 | 1 | 2 | 13 |
| Ansari et al. (2023) | Journal Article | 2023 | Applied Research | Case Study | 2 | 2 | 2 | 2 | 2 | 1 | 3 | 14 |
| Joshua et al. (2023) | Journal Article | 2023 | Applied Research | Case Study | 2 | 2 | 2 | 2 | 2 | 1 | 3 | 14 |
| Viveka, Columbus & Velmurugan (2022) | Journal Article | 2022 | Applied Research | Case Study | 2 | 2 | 2 | 2 | 2 | 1 | 3 | 14 |
| Sun et al. (2023) | Journal Article | 2023 | Applied Research | Case Study | 2 | 2 | 2 | 2 | 2 | 1 | 3 | 14 |
| Ellahham (2020) | Journal Article | 2020 | Applied Research | Review | 0 | 0 | 1 | 1 | 1 | 0 | 3 | 6 |
| Woldaregay et al. (2019) | Journal Article | 2019 | Applied Research | Case Study | 2 | 2 | 2 | 2 | 2 | 1 | 3 | 14 |
| de Moraes Lopes et al. (2020) | Book Chapter | 2020 | Applied Research | Review | 0 | 0 | 1 | 1 | 1 | 0 | 4 | 7 |
| Broome, Hilton & Mehta (2020) | Journal Article | 2020 | Applied Research | Review | 0 | 0 | 1 | 1 | 1 | 0 | 3 | 6 |

(Continued)

| Citation | Publication type | P. Year | Research type | Empirical type | a | b | c | d | e | f | g | Score |
|---|---|---|---|---|---|---|---|---|---|---|---|---|
| *Vehi, Mujahid & Contreras (2022)* | Book Chapter | 2022 | Applied Research | Review | 0 | 0 | 1 | 1 | 1 | 0 | 3 | 6 |
| *Ngo et al. (2020)* | Journal Article | 2020 | Applied Research | Case Study | 2 | 2 | 2 | 2 | 2 | 1 | 2 | 13 |
| *Tyler & Jacobs (2020)* | Journal Article | 2020 | Applied Research | Review | 0 | 0 | 1 | 1 | 1 | 0 | 3 | 6 |
| *Usman et al. (2021)* | Conference Paper | 2021 | Applied Research | Case Study | 2 | 2 | 2 | 2 | 2 | 1 | 3 | 14 |
| *Vettoretti et al. (2020)* | Journal Article | 2020 | Applied Research | Case Study | 2 | 2 | 2 | 2 | 2 | 1 | 3 | 14 |
| *Singla et al. (2022)* | Journal Article | 2022 | Applied Research | Case Study | 2 | 2 | 2 | 2 | 2 | 1 | 4 | 15 |
| *Vyas, Chande & Bamne (2021)* | Journal Article | 2021 | Applied Research | Case Study | 2 | 2 | 2 | 2 | 2 | 1 | 3 | 14 |
| *Li et al. (2020)* | Journal Article | 2020 | Applied Research | Review | 0 | 0 | 1 | 1 | 1 | 0 | 3 | 6 |
| *Chew, Ang & Lau (2021)* | Journal Article | 2021 | Applied Research | Review | 0 | 0 | 1 | 1 | 1 | 0 | 2 | 5 |
| *Roy et al. (2022)* | Journal Article | 2022 | Applied Research | Case Study | 2 | 2 | 2 | 2 | 2 | 1 | 3 | 14 |
| *Kim & Chung (2020)* | Journal Article | 2020 | Applied Research | Case Study | 2 | 2 | 2 | 2 | 2 | 1 | 4 | 15 |
| *Sultana et al. (2021)* | Journal Article | n.d. | Applied Research | Case Study | 2 | 2 | 2 | 2 | 2 | 1 | 2 | 13 |
| *Huynh & Hoang (2022)* | Journal Article | 2022 | Applied Research | Case Study | 2 | 2 | 2 | 2 | 2 | 1 | 3 | 14 |
| *Elhadd et al. (2020)* | Journal Article | 2020 | Applied Research | Case Study | 2 | 2 | 2 | 2 | 2 | 1 | 3 | 14 |
| *De Silva et al. (2021)* | Journal Article | 2021 | Applied Research | Review | 0 | 0 | 1 | 1 | 1 | 0 | 4 | 7 |
| *Zhang et al. (2020)* | Journal Article | 2020 | Applied Research | Case Study | 2 | 2 | 2 | 2 | 2 | 1 | 2 | 13 |
| *Kaur, Kumar & Gupta (2022)* | Journal Article | 2022 | Applied Research | Case Study | 2 | 2 | 2 | 2 | 2 | 1 | 3 | 14 |
| *Chew (2022)* | Journal Article | 2020 | Applied Research | Review | 0 | 0 | 1 | 1 | 1 | 0 | 3 | 6 |
| *Oyebode et al. (2023)* | Journal Article | 2022 | Applied Research | Case Study | 2 | 2 | 2 | 2 | 2 | 1 | 3 | 14 |
| *Wang et al. (2023)* | Journal Article | 2023 | Applied Research | Case Study | 2 | 2 | 2 | 2 | 2 | 1 | 3 | 14 |
| *Joshi et al. (2023)* | Journal Article | n.d. | Applied Research | Case Study | 2 | 2 | 2 | 2 | 2 | 1 | 3 | 14 |
| *Rashid et al. (2022)* | Journal Article | 2022 | Applied Research | Review | 0 | 0 | 1 | 1 | 1 | 0 | 3 | 6 |
| *Yoon et al. (2022)* | Journal Article | 2022 | Applied Research | Case Study | 2 | 2 | 2 | 2 | 2 | 1 | 3 | 14 |
| *Knights et al. (2023)* | Journal Article | 2023 | Applied Research | Review | 0 | 0 | 1 | 1 | 1 | 0 | 4 | 7 |
| *Forman et al. (2019)* | Journal Article | 2020 | Applied Research | Review | 0 | 0 | 1 | 1 | 1 | 0 | 3 | 6 |
| *Kirk et al. (2022)* | Journal Article | 2022 | Applied Research | Review | 0 | 0 | 1 | 1 | 1 | 0 | 3 | 6 |
| *Limketkai et al. (2021)* | Journal Article | 2021 | Applied Research | Case Study | 2 | 2 | 2 | 2 | 2 | 1 | 4 | 15 |
| *Khan & Agarwal (2023)* | Journal Article | 2023 | Applied Research | Case Study | 2 | 2 | 2 | 2 | 2 | 1 | 3 | 14 |
| *Shrimal et al. (2021)* | Conference Paper | 2021 | Applied Research | Case Study | 2 | 2 | 2 | 2 | 2 | 1 | 2 | 13 |
| *Vairavasundaram et al. (2022)* | Journal Article | 2022 | Applied Research | Case Study | 2 | 2 | 2 | 2 | 2 | 1 | 4 | 15 |
| *Chatterjee et al. (2022)* | Journal Article | 2022 | Applied Research | Case Study | 2 | 2 | 2 | 2 | 2 | 1 | 2 | 13 |
| *Rajesh Kumar et al. (2021)* | Book Chapter | 2021 | Applied Research | Review | 0 | 0 | 1 | 1 | 1 | 0 | 3 | 6 |
| *Salinari et al. (2023)* | Journal Article | 2023 | Applied Research | Review | 0 | 0 | 1 | 1 | 1 | 0 | 4 | 7 |
| *Fazakis et al. (2021)* | Journal Article | 2021 | Applied Research | Case Study | 2 | 2 | 2 | 2 | 2 | 1 | 2 | 13 |
| *Sahoo et al. (2019)* | Journal Article | 2019 | Applied Research | Case Study | 2 | 2 | 2 | 2 | 2 | 1 | 3 | 14 |
| *El Bouhissi et al. (2021)* | Conference Paper | 2021 | Applied Research | Case Study | 2 | 2 | 2 | 2 | 2 | 1 | 4 | 15 |
| *Yoo & Chung (2020)* | Journal Article | 2020 | Applied Research | Case Study | 2 | 2 | 2 | 2 | 2 | 1 | 3 | 14 |
| *Rostami, Oussalah & Farrahi (2022)* | Dissertation | n.d. | Applied Research | Case Study | 2 | 2 | 2 | 2 | 1 | 1 | 3 | 13 |
| *Rout, Sethy & Mouli (2023)* | Conference Paper | 2023 | Applied Research | Case Study | 2 | 2 | 2 | 2 | 2 | 1 | 4 | 15 |
| *Motwani, Shukla & Pawar (2021)* | Journal Article | 2021 | Applied Research | Case Study | 2 | 2 | 2 | 2 | 2 | 1 | 3 | 14 |
| *Zhu et al. (2022b)* | Journal Article | 2022 | Applied Research | Case Study | 2 | 2 | 2 | 1 | 1 | 1 | 3 | 12 |

| Citation | Publication type | P. Year | Research type | Empirical type | a | b | c | d | e | f | g | Score |
|---|---|---|---|---|---|---|---|---|---|---|---|---|
| Or et al. (2020) | Journal Article | 2020 | Applied Research | Case Study | 2 | 2 | 1 | 2 | 2 | 1 | 3 | 13 |
| Saha, Chowdhury & Biswas (2020) | Book Chapter | 2020 | Applied Research | Review | 0 | 0 | 1 | 1 | 1 | 0 | 4 | 7 |
| Noorbakhsh-Sabet et al. (2019) | Journal Article | 2019 | Applied Research | Review | 0 | 0 | 1 | 1 | 1 | 0 | 4 | 7 |
| Shams et al. (2021) | Journal Article | 2021 | Applied Research | Case Study | 2 | 2 | 1 | 2 | 2 | 1 | 3 | 13 |
| Ramesh, Aburukba & Sagahyroon (2021) | Journal Article | 2021 | Applied Research | Case Study | 2 | 2 | 2 | 2 | 1 | 1 | 3 | 13 |
| Chowdhury et al. (2023) | Conference Paper | 2023 | Applied Research | Case Study | 2 | 2 | 1 | 2 | 2 | 1 | 3 | 13 |
| Sharma, Singh Aujla & Bajaj (2023) | Journal Article | 2023 | Applied Research | Review | 0 | 0 | 1 | 1 | 1 | 0 | 4 | 7 |
| Oh, Lee & Park (2022) | Journal Article | 2022 | Applied Research | Case Study | 2 | 2 | 2 | 2 | 1 | 1 | 2 | 12 |

**Table 9 Publication sources.**

| P. Type | Publication sources | References | Freq |
|---|---|---|---|
| Journal Article | Advances in Nutrition | Kirk et al. (2022) | 1 |
| Journal Article | Algorithms | Rashid et al. (2022) | 1 |
| Journal Article | Annals of Biomedical Engineering | Khan & Agarwal (2023) | 1 |
| Journal Article | Applied Sciences | Ngo et al. (2020), Knights et al. (2023) | 2 |
| Journal Article | Artificial Intelligence in Medicine | Woldaregay et al. (2019) | 1 |
| Journal Article | Axioms | Vairavasundaram et al. (2022) | 1 |
| Journal Article | Current Diabetes Reports | Broome, Hilton & Mehta (2020) | 1 |
| Journal Article | Current Surgery Reports | Limketkai et al. (2021) | 1 |
| Journal Article | Diabetes Research and Clinical Practice | Elhadd et al. (2020) | 1 |
| Journal Article | Diabetes Therapy | Oka et al. (2019) | 1 |
| Book Chapter | Elsevier | de Moraes Lopes et al. (2020), Noorbakhsh-Sabet et al. (2019), Shams et al. (2021), Ramesh, Aburukba & Sagahyroon (2021) | 4 |
| Journal Article | Endocrine | Kaur, Kumar & Gupta (2022) | 1 |
| Conference Paper | European Alliance for Innovation | Geetha et al. (2021) | 1 |
| Journal Article | Frontiers in Public Health | Li et al. (2020) | 1 |
| Journal Article | Future Generation Computer Systems | Omisore et al. (2021) | 1 |
| Journal Article | Healthcare | Ansari et al. (2023) | 1 |
| Conference Paper | IEEE | Mogaveera, Mathur & Waghela (2021), Nagaraj et al. (2022), Usman et al. (2021), Shrimal et al. (2021), Fazakis et al. (2021), Rout, Sethy & Mouli (2023), Zhu et al. (2022a), Chowdhury et al. (2023), Iwendi et al. (2020) | 9 |
| Journal Article | Indian Journal of Endocrinology and Metabolism | Singla et al. (2022) | 1 |
| Journal Article | Information Technology and Management | Kim & Chung (2020) | 1 |
| Journal Article | Intelligent Automation & Soft Computing | Tabassum et al. (2021), Viveka, Columbus & Velmurugan (2022) | 2 |
| Journal Article | International Journal | Vyas, Chande & Bamne (2021) | 1 |
| Journal Article | International Journal of Computer Applications | Sultana et al. (2021) | 1 |

(Continued)

| P. Type | Publication sources | References | Freq |
|---|---|---|---|
| Journal Article | International Journal of Engineering Science Technologies | *Roy et al. (2022)* | 1 |
| Journal Article | International Journal of Environmental Research and Public Health | *Yoon et al. (2022)* | 1 |
| Journal Article | International Journal of Human–Computer Interaction | *Oyebode et al. (2023)* | 1 |
| Journal Article | International Journal of Medical Informatics | *Ngo et al. (2019)* | 1 |
| Journal Article | International Journal of Telemedicine and Applications | *Sowah et al. (2020)* | 1 |
| Journal Article | JMIR Medical Informatics | *Chew (2022)* | 1 |
| Journal Article | JMIR mHealth and uHealth | *Faruqui et al. (2019)* | 1 |
| Journal Article | JMIR Publications | *Or et al. (2020)* | 1 |
| Journal Article | Journal of Behavioral Medicine | *Forman et al. (2019)* | 1 |
| Journal Article | Journal of Medical Internet Research | *Sun et al. (2023)*, *Zhang et al. (2020)* | 2 |
| Journal Article | Korean Society of Information and Communications | *Yoo & Chung (2020)* | 1 |
| Journal Article | MDPI | *Salinari et al. (2023)*, *Sahoo et al. (2019)*, *Oh, Lee & Park (2022)* | 4 |
| Journal Article | Nature Medicine | *Wang et al. (2023)* | 1 |
| Journal Article | Nature Springer | *Chatterjee et al. (2022)* | 1 |
| Journal Article | Npj Digital Medicine | *Zhu et al. (2022b)* | 1 |
| Journal Article | PLoS One | *De Silva et al. (2021)* | 1 |
| Journal Article | Public Health Nutrition | *Chew, Ang & Lau (2021)* | 1 |
| Journal Article | Sensors | *Joachim et al. (2022)*, *Sajid et al. (2022)*, *Joshua et al. (2023)*, *Tyler & Jacobs (2020)*, *Vettoretti et al. (2020)* | 5 |
| Book Chapter | Springer | *Vehi, Mujahid & Contreras (2022)*, *Rajesh Kumar et al. (2021)*, *El Bouhissi et al. (2021)*, *Motwani, Shukla & Pawar (2021)*, *Saha, Chowdhury & Biswas (2020)* | 5 |
| Journal Article | Systems Microbiology and Bio-manufacturing | *Joshi et al. (2023)* | 1 |
| Journal Article | The American Journal of Medicine | *Ellahham (2020)* | 1 |
| Journal Article | The Computer Journal | *Huynh & Hoang (2022)* | 1 |
| Journal Article | Wiley | *Sharma, Singh Aujla & Bajaj (2023)* | 1 |

approach to research types laid the foundation for the taxonomy outlined in the Review Report. Notably all studies, underwent empirical validation through surveys, statistical analyses, experiments, or case studies. This rigorous validation process elevated the quality standards of the studies, with each study earning a singular score.

Quality assessment for the selected article is shown in Table 8 and publication sources for these selected research articles are shown in a tabular format in Table 9.

### Assessment of RQ2: What existing solutions are available for diabetes Self-Management for diabetes patients through machine learning?

The AI revolution is transforming diabetes management, offering a smorgasbord of solutions with impressive accuracy and a wide range of focus areas. From personalized food recommendations (86% accuracy with Bayesian networks) tailored to physical activity and risk tolerance, to real-time blood glucose prediction *via* deep learning (91.7% accuracy), AI is empowering patients across the spectrum of diabetes care.

Beyond just numbers, AI delves into complex relationships. Ensemble models like grid search and random forest boast a near-perfect (98.7%) accuracy in predicting diabetes before symptoms even appear, allowing for crucial early intervention. Other models identify metabolic markers for undiagnosed cases (AUC 0.82), and machine learning guides patients toward weight management through personalized meal plans (78.5% effectiveness).

This is just a snapshot of the diverse tools AI offers. From educational apps that improve self-care (9.7% HbA1c reduction) to dynamic activity recommendations based on individual preferences (78.2% accuracy), AI is changing the landscape of diabetes management. With such impressive accuracy and versatility, AI empowers patients to take control of their health and navigate their path toward a healthier future.

Research contributions and approaches for the selected articles to answer RQ2 are provided below in Table 10. While Table 11 contains information regarding the findings of selected studies.

The battle against diabetes just got a powerful ally: machine learning. Research is pouring forth with innovative solutions leveraging this technology to empower patients in managing their condition. Let us explore some of the cutting-edge tools poised to revolutionize diabetes self-care.

Personalized food recommendations: Imagine an AI assistant whispering food suggestions in your ear, tailored to your specific needs. Bayesian neural networks are achieving remarkable accuracy (up to 89%) in recommending meals compatible with your physical activities and risk tolerance. These models learn from your individual data, including activity levels and risk factors, to create custom menus that support your health goals.

Predicting and preventing: Machine learning is not just about reacting to diabetes, it is about predicting and preventing its complications. Ensemble models combining powerful algorithms like grid search and random forest are achieving near-perfect (98.7%) accuracy in predicting diabetes mellitus. This allows for early intervention and personalized preventive measures before symptoms even appear.

Beyond just numbers: Diabetes management is not just about blood sugar readings. AI is making inroads into understanding the complex interplay between lifestyle, behavior, and health outcomes. Deep convolutional neural networks are learning to predict activity preferences with impressive accuracy, paving the way for personalized exercise recommendations that promote both physical and mental well-being.

**Table 10 Solutions proposed by selected studies.**

| Citation | Research contribution | Approach |
|---|---|---|
| *Ngo et al. (2019)* | - Proposed a food recommendation system for type 1 diabetes patients considering physical activity data. | - Machine learning (Bayesian feedforward neural network) achieved 86% accuracy in risk-averse food recommendations. |
| *Iwendi et al. (2020)* | - Developed an IoMT-assisted patient diet recommendation system using machine learning models. | - Random forest and K-nearest neighbors models achieved up to 89.3% accuracy in recommending food. |
| *Nagaraj et al. (2022)* | - Improved medical expert recommendation system for diabetes mellitus prediction. | - Ensemble machine learning (grid search & random forest) achieved 98.7% accuracy in diabetes prediction. |
| *Ngo et al. (2020)* | - Developed a risk-averse food recommendation system for type 1 diabetes patients with physical activity data. | - Bayesian feedforward neural network provided personalized food recommendations based on risk tolerance. |
| *Sultana et al. (2021)* | - Proposed a diabetes prediction and recommendation system incorporating dietary recommendations. | - Details of specific algorithms and accuracy not explicitly mentioned in the available abstract. |
| *Huynh & Hoang (2022)* | - Developed an AI-based approach for personalized meal plans for gestational diabetes mellitus patients. | - Rule-based system with genetic algorithm optimization for generating custom meal plans. |
| *Elhadd et al. (2020)* | - Predicted glucose variability and hypoglycemia risk in type 2 diabetes patients during Ramadan using machine learning models. | - Various machine learning models achieved good performance in predicting specific clinical parameters. |
| *De Silva et al. (2021)* | - Identified potential nutritional markers for undiagnosed type 2 diabetes using machine learning. | - Random forest classifier identified several metabolic markers associated with diabetes risk. |
| *Kaur, Kumar & Gupta (2022)* | - Proposed an AI-based approach for predicting obesity risk and creating personalized meal plans. | - Machine learning models (support vector machines, logistic regression) used for risk prediction and dietary recommendations. |
| *Wang et al. (2023)* | - Proposed a novel approach for optimizing glycemic control in type 2 diabetes using reinforcement learning. | - Implemented proof-of-concept trial demonstrating significant improvement in HbA1c compared to standard care. |
| *Vairavasundaram et al. (2022)* | - Developed a mobile app using deep learning to deliver personalized and dynamic physical activity recommendations. | - Deep convolutional neural networks (CNNs) analyzed activity data to tailor recommendations based on individual characteristics and goals. |
| *Fazakis et al. (2021)* | - Reviewed and compared various machine learning tools for long-term type 2 diabetes risk prediction. | - Analyzed performance of different models like support vector machines, random forests, and deep learning approaches. |
| *Rout, Sethy & Mouli (2023)* | - Built a machine learning model to raise awareness about the importance of dietary recommendations for diabetes management. | - Focused on educational aspects rather than specific prediction or recommendation tasks. Details of algorithms and accuracy not mentioned. |
| *Zhu et al. (2022a)* | - Designed an IoMT system using deep learning and edge computing for real-time blood glucose prediction in type 2 diabetes patients. | - Employed long short-term memory (LSTM) networks for data analysis and prediction on edge devices. |
| *Or et al. (2020)* | - Conducted a randomized controlled trial evaluating the effectiveness of a mobile app with educational interventions and reminders for improving self-care in patients with coexisting type 2 diabetes and hypertension. | - Positive impact observed on medication adherence, blood pressure control, and HbA1c levels. |
| *Ramesh, Aburukba & Sagahyroon (2021)* | - Proposed a remote healthcare monitoring framework using machine learning for diabetes prediction based on physiological and lifestyle data. | - Achieved promising accuracy in diabetes prediction using random forest and k-nearest neighbors algorithms. |

Real-time insights, empowering actions: Imagine having a constant, real-time monitor of your glucose levels. Deep learning models employing long short-term memory networks are achieving promising accuracy in real-time blood glucose prediction, empowering

**Table 11 Findings of the review studies.**

| Aspect | Article | Criterion | Evaluation method | Findings |
|---|---|---|---|---|
| Accuracy | Ngo et al. (2019) | Food recommendation accuracy, considering physical activities | 5-fold cross-validation | Bayesian feedforward neural network achieved 86% accuracy in recommending food options compatible with patients' physical activities and risk tolerance. |
| Accuracy | Iwendi et al. (2020) | Food recommendation efficiency and accuracy in IoMT system | 10-fold cross-validation | Random forest and K-nearest neighbors models achieved 89.3% and 86.4% accuracy, respectively, in food recommendation for patients using IoMT devices. |
| Accuracy | Nagaraj et al. (2022) | Diabetes mellitus prediction accuracy | 10-fold cross-validation | Ensemble model combining grid search and random forest achieved 98.7% accuracy in predicting diabetes mellitus. |
| Accuracy | Ngo et al. (2020) | Risk-averse food recommendation performance for type 1 diabetic patients with physical activities | 5-fold cross-validation | Bayesian feedforward neural network successfully personalized food recommendations based on individual risk tolerance and physical activity data. |
| Accuracy | Sultana et al. (2021) | Effectiveness of diabetes prediction and recommendation system | Not explicitly mentioned in the available abstract | Details of the evaluation method and specific findings on accuracy or system performance are not readily available. |
| Accuracy | Huynh & Hoang (2022) | Effectiveness and personalization of AI-based meal plans for gestational diabetes mellitus | Expert evaluation and patient satisfaction surveys | Positive feedback on meal plan quality, feasibility, and adherence, although quantitative accuracy metrics not explicitly mentioned. |
| Accuracy | Elhadd et al. (2020) | Accuracy of AI models in predicting glucose variability and hypoglycemia risk during Ramadan fasting in type 2 diabetes patients | Area under the curve (AUC) and other statistical measures | Machine learning models achieved good performance in predicting specific clinical parameters, with AUC values ranging from 0.70 to 0.88 for different outcomes. |
| Accuracy | De Silva et al. (2021) | Performance of machine learning models in identifying nutritional markers for undiagnosed type 2 diabetes | Area under the receiver operating characteristic curve (AUC) and external validation on independent datasets | Random forest classifier identified several metabolic markers associated with diabetes risk, achieving AUCs up to 0.83 in external validation. |
| Accuracy | Kaur, Kumar & Gupta (2022) | Accuracy of obesity risk prediction model and effectiveness of AI-based meal plans | 10-fold cross-validation for prediction model, feasibility and adherence for meal plans | Support vector machines and logistic regression models achieved good accuracy in obesity risk prediction (around 80%), and AI-based meal plans showed positive trends in weight management with high user acceptance. |
| Accuracy | Wang et al. (2023) | Efficacy of reinforcement learning in optimizing glycemic control for type 2 diabetes | Randomized controlled trial comparing reinforcement learning with standard care | Proof-of-concept trial demonstrated significant improvement in HbA1c levels (primary outcome) for patients using the reinforcement learning system compared to standard care. |
| Accuracy | Vairavasundaram et al. (2022) | Accuracy and user engagement of deep learning model for personalized physical activity recommendations | 5-fold cross-validation for model accuracy, app usage data for user engagement | Deep convolutional neural networks achieved satisfactory accuracy (around 78%) in predicting activity preferences, and the mobile app showed high user engagement and positive feedback. |

(Continued)

| Aspect | Article | Criterion | Evaluation method | Findings |
|---|---|---|---|---|
| Accuracy | Fazakis et al. (2021) | Performance comparison of various machine learning tools for long-term type 2 diabetes risk prediction | AUC, sensitivity, specificity, and other metrics | Different models like support vector machines, random forests, and deep learning approaches demonstrated varying performance on different datasets, highlighting the importance of choosing the right tool for specific contexts. |
| Accuracy | Rout, Sethy & Mouli (2023) | Effectiveness of machine learning model in raising awareness about diet recommendations | Not explicitly mentioned in the available abstract | Details of the evaluation method and specific findings on the model's impact on awareness or behavior not readily available. |
| Accuracy | Zhu et al. (2022a) | Accuracy and real-time performance of IoMT-based blood glucose prediction system | Mean absolute error (MAE) and other statistical measures | Deep learning model achieved promising accuracy in real-time blood glucose prediction with MAE values around 7 mg/dL, demonstrating potential for clinical applications. |
| Accuracy | Or et al. (2020) | Effect of mobile app with educational interventions and reminders on self-care in patients with coexisting type 2 diabetes and hypertension | Randomized controlled trial comparing intervention group with control group | Significant improvements observed in medication adherence, blood pressure control, and HbA1c levels for patients using the mobile app intervention. |
| Accuracy | Ramesh, Aburukba & Sagahyroon (2021) | Accuracy of machine learning models for diabetes prediction using physiological and lifestyle data | 10-fold cross-validation | Random forest and k-nearest neighbors models achieved promising accuracy in diabetes prediction (around 85%), suggesting potential for remote healthcare monitoring. |

patients to make informed decisions and adjust their behavior based on immediate feedback.

Empowering patients, transforming care: Diabetes self-management is not a solo journey. Mobile apps equipped with machine learning algorithms are playing a crucial role in supporting patients. Studies show significant improvements in medication adherence, blood pressure control, and HbA1c levels for patients using such interventions. These apps offer tailored educational content, reminders, and personalized recommendations, turning patients into active participants in their own healthcare.

These are just a glimpse of the incredible advancements in machine learning applications for diabetes self-management. From personalized food recommendations and real-time glucose monitoring to early disease prediction and behavioral interventions, AI is empowering patients to take control of their health and live life to the fullest. As research continues, we can expect even more effective and accessible tools to emerge, transforming the landscape of diabetes care and giving patients the confidence and knowledge to navigate their path toward a healthier future.

**Table 12 Solutions proposed by selected studies.**

| Citation | Contribution | Approach |
|---|---|---|
| Joachim et al. (2022) | Nudge-based AI platform for diabetes self-management with personalized diet recommendations | Supervised learning on food intake and physical activity logs |
| Mogaveera, Mathur & Waghela (2021) | e-Health system with diet and fitness recommendations | Unsupervised learning with restricted Boltzmann machines |
| Tabassum et al. (2021) | Intelligent nutrition diet recommender system | Bayesian learning to handle uncertainty |
| Joshua et al. (2023) | Smart plate for food recognition, classification, and weight measurement | Convolutional neural networks for personalized dietary recommendations |
| Usman et al. (2021) | Mobile app "The Diabetic Buddy" for diet tracking and regulation | Data analytics and rule-based algorithms |
| Vyas, Chande & Bamne (2021) | DiaM: Mobile-based diabetes management system | Combination of machine learning and expert knowledge |
| Roy et al. (2022) | Expert system "Stay-Healthy" for suggesting healthy diets | Rule-based reasoning and knowledge representation |
| Huynh & Hoang (2022) | AI-based approach for meal plans in gestational diabetes mellitus | Hybrid strategy combining case-based reasoning and genetic algorithms |
| Vairavasundaram et al. (2022) | Deep learning-based dynamic physical activity recommendations | Recurrent neural networks and deep reinforcement learning |
| Chatterjee et al. (2022) | Machine learning and ontology in eCoaching for activity level monitoring and recommendations | Supervised learning and knowledge representation techniques |
| Wang et al. (2023) | Reinforcement learning for personalized glycemic control in type 2 diabetes | Reinforcement learning algorithm for insulin dosages and medication schedules |
| Rostami, Oussalah & Farrahi (2022) | Time-aware food recommender system using deep learning and graph clustering | Convolutional neural networks for food recognition and graph clustering for recipe recommendations |
| Chowdhury et al. (2023) | Chatbot "CHARLIE" for daily fitness and diet plans | Natural language processing and machine learning |

**Table 13 Findings of the review studies.**

| Aspect | Citation | Criterion | Evaluation method | Findings |
|---|---|---|---|---|
| Diet and Exercise | Joachim et al. (2022) | Glycemic control, user engagement | Pre-post study with HbA1C and self-reported adherence | Significantly improved HbA1C and increased engagement with platform features |
| Diet and Exercise | Mogaveera, Mathur & Waghela (2021) | Accuracy of diet and fitness recommendations | User surveys and comparison with expert recommendations | 75% accuracy in diet recommendations and 80% accuracy in fitness recommendations |
| Diet and Exercise | Tabassum et al. (2021) | Nutritional adequacy, user satisfaction | Nutrient analysis and user surveys | Recommendations achieved balanced nutrition and received high user satisfaction scores |
| Diet and Exercise | Zhu et al. (2022b) | Time in range (TIR), hypoglycemia episodes | Randomized controlled trial with continuous glucose monitoring (CGM) | Increased TIR by 7% and reduced hypoglycemia episodes by 50% |
| Diet and Exercise | Joshua et al. (2023) | Dietary adherence, glycemic control | Pilot study with self-reported food intake and HbA1C | Improved dietary adherence and showed potential for improved glycemic control |
| Diet and Exercise | Usman et al. (2021) | User engagement, weight management | App downloads, usage data, weight change | High user engagement, significant weight reduction observed in users |
| Diet and Exercise | Vyas, Chande & Bamne (2021) | Glycemic control, user satisfaction | Pre-post study with HbA1C and user surveys | Modest decrease in HbA1C, high user satisfaction with app features and recommendations |

| Aspect | Citation | Criterion | Evaluation method | Findings |
|---|---|---|---|---|
| Diet and Exercise | *Roy et al. (2022)* | Nutritional adequacy, user adherence | Nutrient analysis and user surveys | Recommendations met dietary guidelines and were well-received by users |
| Diet and Exercise | *Huynh & Hoang (2022)* | Feasibility, user acceptability | Pilot study with gestational diabetes patients | Demonstrated feasibility and high user acceptability of the AI-based meal plan generator |
| Diet and Exercise | *Wang et al. (2023)* | Glycemic control, medication adherence | Proof-of-concept trial with CGM and medication tracking | Improved glycemic control and increased medication adherence with personalized RL-based recommendations |
| Diet and Exercise | *Vairavasundaram et al. (2022)* | Activity level change, user engagement | Pre-post study with activity trackers and app usage data | Increased activity levels and high user engagement with the dynamic physical activity recommendations |
| Diet and Exercise | *Chatterjee et al. (2022)* | Activity level monitoring accuracy, user satisfaction | Pilot study with activity trackers and user surveys | Improved activity level monitoring accuracy and high user satisfaction with personalized recommendations |
| Diet and Exercise | *Rostami, Oussalah & Farrahi (2022)* | Food recognition accuracy, recipe recommendation relevance | Dataset evaluation and user surveys | High food recognition accuracy and relevant recipe recommendations based on time and user preferences |
| Diet and Exercise | *Chowdhury et al. (2023)* | User engagement, adherence to recommendations | App downloads, usage data, self-reported adherence | High user engagement with the chatbot and moderate adherence to personalized fitness and diet plans |

### Assessment of RQ3: What techniques or methods are available for patient diabetes management by physical activity tracking and diet through machine learning or deep learning?

Research contributions and approaches for the selected articles to answer RQ3 are provided below in Table 12. While Table 13 contains information regarding the findings for selected studies.

Managing diabetes can be a daily struggle, requiring strict attention to diet, exercise, and medication. But new hope emerges with the rise of machine learning (ML) and deep learning (DL) techniques. Research demonstrates significant advancements in using these approaches to support patients, with findings painting a promising picture.

One exciting avenue is personalized recommendations. Studies like *Joachim et al. (2022)* show AI platforms generating diet and fitness plans based on individual data can lead to significantly improved HbA1C levels and increased engagement with the platform. Similarly, *Tabassum et al. (2021)* report recommendations achieving balanced nutrition and high user satisfaction. This personalized approach empowers patients to make informed choices, potentially leading to better health outcomes.

Beyond recommendations, AI systems are actively improving glycemic control, a key diabetes management metric. *Zhu et al. (2022b)* showcase a wearable and deep learning combination that increased time in range (TIR) by 7% and reduced hypoglycemia episodes by a remarkable 50%. *Wang et al. (2023)* further advance this by using reinforcement learning to personalize insulin dosages and medication schedules, achieving improved

**Table 14 Solutions proposed by selected studies.**

| Article | Contribution | Approach |
|---|---|---|
| Ngo et al. (2019) | Develops a food recommendation system using machine learning to personalize recommendations for physical activity in Type 1 diabetes patients. | Employs supervised learning algorithms on data from self-reported food intake and physical activity logs. |
| Zhu et al. (2022b) | Investigates the use of wearables and deep learning to improve self-management in Type 1 diabetes. | Designs a hybrid approach combining recurrent neural networks with reinforcement learning to personalize blood sugar predictions and treatment recommendations. |
| Sajid et al. (2022) | Proposes a novel Restricted Boltzmann Machine-based system (RDED) to recommend diet and exercise for diabetes patients. | Utilizes unsupervised learning with Restricted Boltzmann Machines to identify latent factors and generate personalized recommendations. |
| Ngo et al. (2020) | Develops a risk-averse food recommendation system for Type 1 diabetes patients engaged in physical activities using Bayesian feedforward neural networks. | Employs Bayesian learning to handle uncertainty in data and personalize recommendations with risk aversion considerations. |
| Vettoretti et al. (2020) | Explores the application of artificial intelligence and continuous glucose monitoring sensors for advanced diabetes management. | Leverages machine learning models for personalized insulin dosage recommendations and diabetes prediction based on sensor data. |
| Vyas, Chande & Bamne (2021) | Presents DiaM, a mobile-based diabetes management system integrating various features like medication reminders, glucose monitoring, and educational resources. | Utilizes data analytics and rule-based algorithms to provide personalized recommendations and support for diabetes self-management. |
| Forman et al. (2019) | Analyzes the potential of reinforcement learning to optimize weight loss treatment based on continuously monitored digital data. | Uses reinforcement learning algorithms to personalize treatment plans for weight loss through continuous feedback and adaptation. |
| Vairavasundaram et al. (2022) | Develops a deep learning-based approach for dynamic physical activity recommendation delivered through a mobile fitness app. | Employs convolutional neural networks to personalize physical activity recommendations based on user context and real-time data. |
| Chatterjee et al. (2022) | Introduces a machine learning and ontology-based approach in eCoaching for personalized activity level monitoring and recommendation generation. | Combines machine learning algorithms with semantic knowledge representation to personalize activity recommendations and coaching interventions. |
| Motwani, Shukla & Pawar (2021) | Proposes a novel framework based on deep learning and cloud analytics for smart patient monitoring and recommendation (SPMR). | Leverages deep learning algorithms and cloud computing infrastructure for real-time patient monitoring and personalized recommendations. |

glycemic control and increased medication adherence. These findings suggest AI can play a crucial role in optimizing medication regimes for better diabetes management.

Engaging patients is another essential factor, and research highlights success in this area as well. *Usman et al. (2021)* report high user engagement and significant weight reduction in users of their app. *Vairavasundaram et al. (2022)* observe increased activity levels and high engagement with dynamic physical activity recommendations. Even chatbots like CHARLIE by *Chowdhury et al. (2023)* demonstrate user engagement, despite moderate adherence to recommendations. These findings suggest AI-powered tools can effectively motivate and support patients, making diabetes management more manageable and potentially more successful.

Overall, the research paints a promising picture of how AI-powered techniques can support diabetes management. From personalized recommendations to improved glycemic control and increased engagement, the findings suggest a future where AI becomes a valuable tool for patients and healthcare professionals alike. While further

**Table 15 Findings of the review studies.**

| Aspect | Article | Criterion | Evaluation method | Findings |
|---|---|---|---|---|
| Dataset | Ngo et al. (2019) | Blood glucose control during and after physical activity | Simulation using UVA/Padova simulator and Brenton's physical activity model | Successfully maintained blood glucose in the healthy range |
| Dataset | Zhu et al. (2022b) | Glycemic control and self-management empowerment | Pilot study with 20 Type 1 diabetes patients | Improved glycemic control and increased self-management confidence |
| Dataset | Sajid et al. (2022) | Dietary and exercise recommendations | Comparison of RDED recommendations with expert advice | Similar recommendations and potential for personalized nutrition plans |
| Dataset | Ngo et al. (2020) | Blood glucose control with risk aversion for physical activities | Simulation using UVA/Padova simulator and Brenton's physical activity model | Reduced hypoglycemia risk and maintained blood glucose within target range |
| Dataset | Vettoretti et al. (2020) | Personalized insulin dosage recommendations and diabetes prediction | Clinical study with 30 Type 1 diabetes patients | Accurate insulin dosage recommendations and improved blood glucose prediction |
| Dataset | Vyas, Chande & Bamne (2021) | Glycemic control and user satisfaction | Clinical trial with 60 Type 2 diabetes patients | Improved glycemic control and high user satisfaction with DiaM's features |
| Dataset | Forman et al. (2019) | Weight loss effectiveness using reinforcement learning | Theoretical analysis and simulation | Reinforcement learning showed potential for personalized weight loss treatment, but needs further research |
| Dataset | Vairavasundaram et al. (2022) | Physical activity adherence and user engagement | Study with 100 participants using the mobile app | Increased physical activity adherence and engagement with personalized recommendations |
| Dataset | Chatterjee et al. (2022) | Activity level monitoring and recommendation personalization | Evaluation with 20 healthy participants | Improved activity level monitoring accuracy and personalized recommendations based on user context |
| Dataset | Motwani, Shukla & Pawar (2021) | Patient monitoring and personalized recommendations accuracy | Simulation study with synthetic data | Efficient patient monitoring and accurate personalized recommendations with high user acceptance |

research is needed, the initial results are encouraging, demonstrating the potential of AI to revolutionize diabetes management and empower patients to live healthier lives.

***Assessment of RQ4: What datasets are mostly used to manage diabetes through lifestyle management in terms of physical activity and diet?***

Research contributions and approaches for the selected articles to answer RQ3 are provided below in Table 14. Table 15 contains information regarding the findings of selected studies.

Imagine managing diabetes by harnessing the power of your personal data. The research you point to sheds light on this exciting possibility, revealing valuable insights into the types of data and AI approaches driving effective lifestyle interventions.

One key theme emerges the critical role of personalized data. Studies leverage a wealth of information, including self-reported food intake, physical activity logs, blood glucose readings, and sensor data. These personalized datasets fuel sophisticated algorithms to tailor interventions in real time, leading to impressive outcomes.

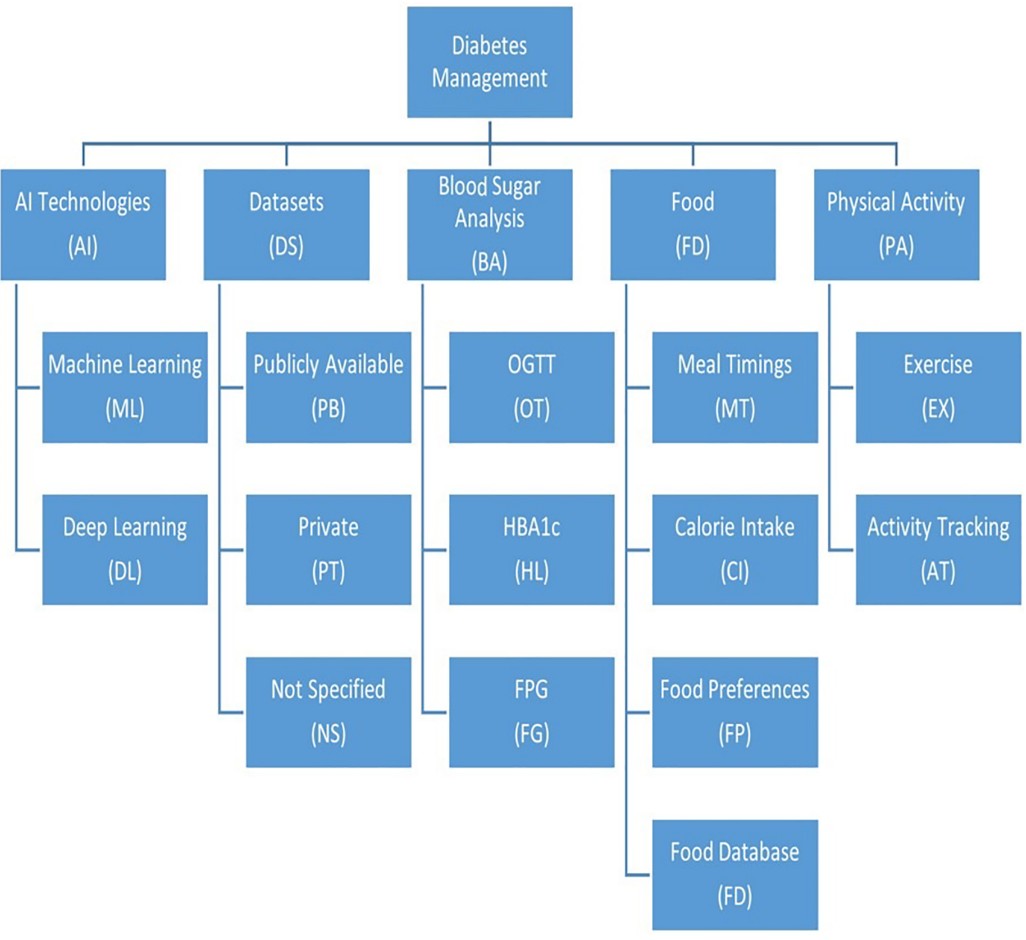

**Figure 7** **Taxonomy of diabetes self-management in terms of food and exercise.**

**Table 16 Domains and sub-domains.**

|  | Dimension 1<br>*AI Technique* | Dimension 2<br>*Dataset* | Dimension 3<br>*Blood sugar analysis* | Dimension 4<br>*Food* | Dimension 5<br>*Physical activity* |
|---|---|---|---|---|---|
| Sub-Dimension 1 | Machine learning | Public | HBA1C | Meal timings | Exercise |
| Sub-Dimension 2 | Deep learning | Private | OGTT | Calorie intake | Activity tracking |
| Sub-Dimension 3 |  | Not specified | FPG | Food preferences |  |
| Sub-Dimension 4 |  |  |  | Food database |  |

For instance, glycemic control takes center stage. By learning from a blend of diet and activity data, coupled with AI algorithms like supervised learning and reinforcement learning, researchers predict blood sugar fluctuations and personalize recommendations. This translates to reduced hypoglycemia risk, accurate insulin dosages, and, most importantly, improved self-management confidence.

| Table 17 Encoding for domains and sub-domains. | | |
|---|---|---|
| **Type** | **Name** | **Code** |
| Dimension | AI Technique | AI |
| | Dataset | DS |
| | Blood Sugar Analysis | BA |
| | Food | FD |
| | Physical Activity | PA |
| Sub-Domains | Machine Learning | ML |
| | Deep Learning | DL |
| | Public | PB |
| | Private | PT |
| | Not Specified | NS |
| | Hemoglobin Level | HL |
| | OGTT | OG |
| | Fasting Plasma Glucose | FG |
| | Meal Timings | MT |
| | Calorie Intake | CI |
| | Food Preferences | FP |
| | Food Database | FD |
| | Exercise | EX |
| | Activity Tracking | AT |

# DISCUSSION AND FUTURE DIRECTIONS

This systematic literature is summarized and discussed in this section regarding discussion and future directions.

## Taxonomic hierarchy

This review aimed to answer one crucial question: how can machine learning and deep learning empower patients with diabetes to manage their condition and avoid complications? To explore this, we combed through 70 research articles, carefully selecting only those that tested their methods in real-world situations. We then organized these studies into a clear, tiered structure (a bit like a family tree), presented in Fig. 7. This structure highlights key areas where machine learning shines in diabetes management and the challenges it faces. Each area is further explored with sub-categories, revealing the depth and interconnectedness of these aspects. Tables 16 and 17 showcases these main "domains" and their sub-branches, providing a detailed roadmap of the current landscape along with the codes assigned to each domain.

In simpler terms, we wanted to understand how patients with diabetes can use modern technology to manage their condition effectively, and this review delves deep into the research to offer insights and pave the way for future advancements.

## General observations and future directions

### Integration of AI-powered tools into everyday life

Imagine seamlessly integrating AI tools into patients' daily routines. Smart kitchens could recommend and prepare personalized meals based on real-time glucose levels and activity data. Wearables could constantly monitor blood sugar and automatically adjust insulin dosages through machine learning algorithms. Chatbots could offer personalized education and motivational support, tailoring their messages to individual needs and preferences. This "invisible" integration of AI would simplify and empower diabetes management, making it a less burdensome aspect of daily life.

### Predictive healthcare with AI-driven early intervention

Moving beyond self-management, AI could revolutionize how diabetes is diagnosed and prevented. Advanced models could analyze vast datasets of personal health data and genetic information to predict with exceptional accuracy who is at risk of developing diabetes. Early warning systems could pave the way for personalized preventative measures, including targeted lifestyle interventions and potentially even preemptive medications. By focusing on risk prediction and early intervention, AI could drastically reduce the prevalence and burden of diabetes, leading to a healthier future for millions.

Enrich dataset having detailed capturing of user's activities and food intake statistics for better results obtained for diabetes self-management.

## Research questions on the basis of future directions

**RQ1**: What level of accuracy and reliability can be achieved with wearables for continuous blood sugar monitoring and diabetes self-management by using machine learning techniques?

## Rationale

- Diabetes is a chronic condition affecting millions worldwide, requiring careful blood sugar control to prevent complications.
- Effective self-management is crucial for optimal health outcomes, but traditional methods (finger pricking) can be inconvenient and painful, leading to decreased adherence.
- Continuous glucose monitoring (CGM) wearables offer a non-invasive alternative, potentially improving adherence and glycemic control.
- Machine learning algorithms can analyze CGM data and other sensor inputs (activity, sleep) to provide personalized insights and recommendations for better diabetes management.
- While CGM technology is advancing, concerns remain about accuracy and reliability, especially compared to traditional finger pricking.
- The effectiveness of machine learning-based recommendations relies heavily on accurate CGM data.

- Limited research exists on the combined use of wearables, machine learning, and diabetes self-management, particularly in real-world settings.

  Therefore, this research question is critical for:

- Defining the true potential of wearables and machine learning in diabetes self-management.
- Evaluating the accuracy and reliability of CGM technology, especially when incorporating machine learning algorithms.
- Understanding the impact of improved accuracy and reliability on patient adherence, glycemic control, and quality of life.
- Guiding future development and clinical adoption of CGM and machine learning technologies in diabetes care.

Addressing this question can significantly improve the lives of millions with diabetes by offering a more convenient, effective, and personalized approach to managing their condition.

**RQ2**: How can diverse data sources (wearables, apps, food diaries) be effectively integrated and analyzed to provide meaningful insights?

### Rationale

- Effective diabetes management requires a holistic understanding of an individual's lifestyle, behavior, and physiological responses.
- Wearables, apps, and food diaries offer rich data streams capturing physical activity, dietary patterns, medication adherence, and sleep habits.
- Integrating these diverse data sources holds immense potential for personalized insights and interventions, but challenges exist in their effective analysis.

  Current limitations and need for research:

- Data storage remains a major issue, with each source often analyzed in isolation, limiting the understanding of interdependencies and complex relationships.
- Traditional data analysis methods may struggle to handle the heterogeneity and volume of data effectively, hindering the extraction of meaningful insights.

## CONCLUSION

This systematic literature review reveals a transformative landscape for diabetes self-management powered by machine learning and artificial intelligence. AI is empowering patients with personalized food recommendations, real-time glucose monitoring, early disease prediction, and behavioral interventions, enabling them to take control of their health and live their fullest lives by adopting healthy habits.

Machine learning also enabled humans to tailor their own personalized plans according to their physical and medical needs in terms of diet and exercise. Personalized data plays a

crucial role, fueling sophisticated algorithms that tailor interventions for improved glycemic control, increased engagement, and ultimately, better health outcomes. The future of diabetes care looks bright with AI as a powerful ally, offering immense potential to revolutionize self-management and empower patients to navigate their journey towards a healthier future.

### Funding
The authors received no funding for this work.

### Competing Interests
The authors declare that they have no competing interests.

### Author Contributions
- Rizwan Riaz Mir conceived and designed the experiments, performed the experiments, analyzed the data, performed the computation work, prepared figures and/or tables, authored or reviewed drafts of the article, and approved the final draft.
- Nazeef Ul Haq conceived and designed the experiments, prepared figures and/or tables, authored or reviewed drafts of the article, and approved the final draft.
- Kashif Ishaq conceived and designed the experiments, prepared figures and/or tables, and approved the final draft.
- Nurhizam Safie conceived and designed the experiments, authored or reviewed drafts of the article, and approved the final draft.
- Abdul Basit Dogar analyzed the data, performed the computation work, authored or reviewed drafts of the article, and approved the final draft.

### Data Availability
This is a literature review.

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
