# Peer review of "Impact of machine learning on dietary and exercise behaviors in type 2 diabetes self-management: a systematic literature review"

_PeerJ Computer Science, doi:10.7717/peerj-cs.2568_

## Round 0.1 · original submission · Major Revisions

Dear authors,

Thank you for submitting your Literature Review article. Feedback from the reviewers is now available. We strongly recommend that you address the issues raised by the reviewers and resubmit your paper after making the necessary changes.

Best wishes,

·

Basic reporting

In the study, an extensive literature review was conducted to link the relationship between diet and exercise. It is generally appropriate in terms of language, but it would be good to review it once again. Sufficient number of articles were examined within the scope of the study and the article structure is academically appropriate. Sufficient tables were used in the study.

Minor adjustments should be made taking into account the following comments

There is no explanation for Table 2 in the text.
It is difficult to understand because the table locations are not fully specified in the text. Although the tables are given as a separate appendix, the location (title) of the table should be indicated.
Some of the figures are out of order. The format of these figures should be edited.

Experimental design

The study is compatible with the scope of the journal. No violation of ethical standards was observed. The research method for the literature review described in the article is appropriate for a survey article. Necessary citations are presented correctly. The survey is presented correctly in terms of logic.

Validity of the findings

The findings of the article are in line with the general flow of the article. I think that the discussion in the Conclusion section for the Survey article should be increased a little more

Additional comments

The Survey article is generally well written, but some minor editing, taking into account the points mentioned above, would improve the overall process of the article.

·

Basic reporting

Strengths:
The paper covers a timely and relevant topic of broad interest.
The introduction provides adequate context and background on diabetes management and machine learning applications - although a probably overly historical introduction to diabetes as a condition ('Diabetes is known as one of the oldest chronic diseases first reported 3000 years back in an
Egyptian manuscript. Araetus of Cappadocia (81-133AD) first used the termî Diabetes')
The literature is well-referenced, with 70 papers reviewed from 2019-2023.

Needs improvement:
The English language usage needs significant improvement throughout. There are numerous grammatical errors and awkward phrasings that make reading difficult. This is the main issue with this manuscript and would require extensive re-writing.
The structure does not fully conform to standard systematic review formats. A clearer delineation of methods, results, and discussion sections would improve readability.
While the scope is appropriate, the paper could better articulate its unique contribution given recent reviews in the field.

Experimental design

Strengths:
The review covers a comprehensive range of topics related to ML/AI in diabetes self-management.
The authors attempt to provide a systematic approach to literature selection and analysis.

Needs improvement:
The survey methodology lacks sufficient detail to ensure replicability. Specific inclusion/exclusion criteria and search strategies should be clearly stated.
The quality assessment process for included studies is not well-described. A standardised quality assessment tool should be used and reported.
The overall organisation of the review could be improved with a more logical grouping of topics and clearer subsections.

Validity of the findings

Strengths:
The review identifies several promising applications of ML/AI in diabetes management.
The authors attempt to synthesise findings across multiple studies.

Needs improvement:
The conclusions are not always clearly linked to the evidence presented. A more structured approach to summarising key findings for each research question would strengthen the paper.
Critical analysis of the quality and limitations of included studies is limited.
The review would benefit from a more logically structured discussion and future studies section - the current section is difficult to follow

Additional comments

While this paper has potential, in my opinion it would require substantial revision before being considered for publication.

Points to focus on:
Improving the clarity and professionalism of the writing <- this is critical
Strengthening the methodological rigor and transparency.
Enhancing the critical analysis and synthesis of findings.
Providing more specific, actionable insights for future research and clinical practice.

---

## Round 0.2 · accepted · Accept

Dear Authors,

I am grateful for your efforts in revising the paper. One of the previous reviewers has not submitted their assessment within the specified timeframe. However, other reviewer has indicated that your revised paper is suitable for acceptance in its current form. I am also satisfied with the revised manuscript and believe it is now ready for publication.

Best wishes,

·

Basic reporting

no comment

Experimental design

no comment

Validity of the findings

no comment

Additional comments

All the arrangements given to me for revision have been made by the authors and their responses have been given. I believe that the article can be accepted in its current state.